# Two types of motifs enhance human recall and generalization of long sequences
Shuchen Wu [1] ✉, Mirko Thalmann[2] & Eric Schulz[2]

Whether it is listening to a piece of music, learning a new language, or solving a mathematical equation, people often acquire abstract notions in the sense of motifs and variables—manifested in musical themes, grammatical categories, or mathematical symbols. How do we create abstract representations of sequences? Are these abstract representations useful for memory recall? In addition to learning transition probabilities, chunking, and tracking ordinal positions, we propose that humans also use abstractions to arrive at efficient representations of sequences. We propose and study two abstraction categories: projectional motifs and variable motifs. Projectional motifs find a common theme underlying distinct sequence instances. Variable motifs contain symbols representing sequence entities that can change. In two sequence recall experiments, we train participants to remember sequences with projectional and variable motifs, respectively, and examine whether motif training benefits the recall of novel sequences sharing the same motif. Our result suggests that training projectional and variables motifs improve transfer recall accuracy, relative to control groups. We show that a model that chunks sequences in an abstract motif space may learn and transfer more efficiently, compared to models that learn chunks or associations on a superficial level. Our study suggests that humans construct efficient sequential memory representations according to the two types of abstraction we propose, and creating these abstractions benefits learning and out-of-distribution generalization. Our study paves the way for a deeper understanding of human abstraction learning and generalization.

When the iconic notes strike: GGGE♭, FFFD,—Beethoven's Fifth Symphony comes immediately to our mind. As the music progresses, we note the change of motif to GGGB or GGGC, variations in forms and voices, one at each step. Our ability to effortlessly identify those forms of abstract motifs endows us with an ability to learn mathematics, languages, and music. From representing "x" as a variable to perceiving 'noun' as a category including "cats", "dogs", and "elephants", these abstract motifs automatically come to our mind and help us to memorize sequences and generalize to novel situations. How do we abstract motifs from perceiving sequences? What advantages does this ability confer in terms of memory representations and transfer? More importantly, how do we construct an abstract representation during learning?

The literature suggests that we have the capability to learn multi-faced aspects of sequences. In what is known as grammatical judgment tasks in artificial grammar learning, after familiarizing participants to a set of grammatically valid sequences generated by a finite state language[1,2], participants acquire the ability to distinguish unseen grammatical from ungrammatical sequences[3,4]. Further research suggested that sequence learning extends beyond learning first-order transition probabilities. Frequently occurring fragments shared between the test and training sequences influence test judgment[5] and are more likely to be judged as grammatical[6–8]. Such phenomenon can be explained by models that learn repeated sequence fragments as chunks[9–12].

Beyond learning sequence fragments and transition probabilities, a few studies suggest the early cognitive capability to acquire sequential patterns on an abstract level: After familiarizing infants early as 7-month-old to sequences such as AAB and CCD, they were likelier to direct their gaze toward novel sequences sharing the same structure, such as DDF, rather than a different structure, such as KTK. Such ability to capture what was named as 'abstract algebraic structure'[13] in sequences cannot be explained by learning transition probabilities or chunks. Meanwhile, another abstract pattern has been hypothesized by linguists: we acquire sequence knowledge on a symbolic level[14–16]. This ability is a prerequisite to learning phase structures on the level of symbols such as noun phrase = determinant +

[1]Max Planck Institute for Biological Cybernetics, Tübingen, Germany. [2]Helmholtz Institute for Human-Centered AI, München, Germany.
✉e-mail: shuchen.wu@tuebingen.mpg.de

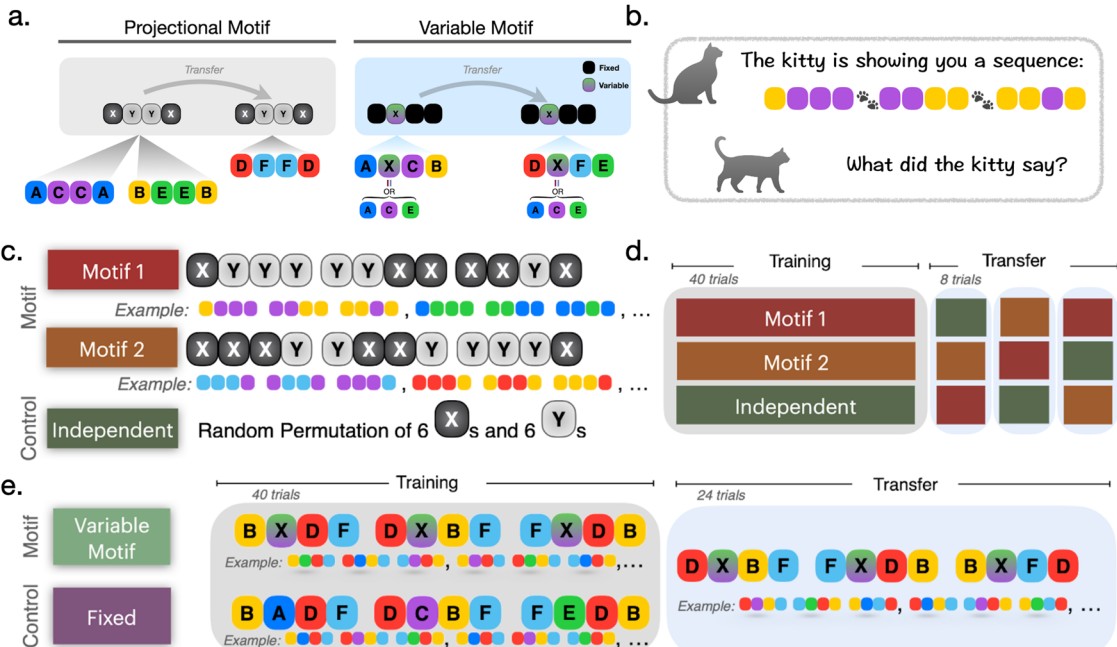

**Fig. 1 | Taxonomy of motifs and experimental design. a** A taxonomy of sequence motifs. Projectional motifs refer to patterns of sequences in a projected space that are mapped from the concrete sequence space by a projection function. In the example being shown, the projection function finds the distinct items in the sequence and maps sequential observation into a binary sequence XYYX, with X being the first unique item appearing and Y being the second. Variable motifs refer to a pattern of invariant (dark box) and variant (gradient-colored box) sequential elements. The variable motif contains at some position a variable–a symbol representing a sequential component that can vary. Such a variable is identified when any of the sequential components it represents is identified. In this example, the variable X represents the possible occurrence of A, C, or E. We hypothesized that participants could learn both types of motifs through practice and exploit their knowledge of both motif types in memorizing and generalizing to novel sequences. **b** We study motif learning in a sequence recall task. Participants are instructed to remember a

sequence of 12 colors. To make the sequence more digestible, the colors are displayed one after another in three groups of four items separated by the display of pair of paws after each group. **c** Experiment 1 studies learning projectional motifs. Participants are divided into three groups. Two motif groups (Motif 1 and Motif 2) and one control group (Independent). **d** Each group is first trained on their respective motif or random sequences (Independent) and then tested on randomly interleaved transfer blocks of three types. There are no overlapping sequences between all transfer blocks and training blocks. **e** Experiment 2 studies learning variable motifs. The variable motif group is trained on sequences with an underlying variable motif. That is, the second position of each subsequence display is randomly drawn among three colors (purple, blue, or green). The fixed group is trained to recall fixed sequences. Both groups are then subsequently tested on novel sequences sharing the variable motif.

noun, and helps us to judge the grammaticity of very unlikely-occurring sentences[17].

In this work, we zoom in, refine, and categorize different forms of abstract sequential structures. We define and differentiate between two algebraic abstractions: "projectional motifs," which are patterns derived from sequences using a projectional function, and "variable motifs," which include patterns involving concrete and variable elements. We move beyond grammaticity judgements and examine the role of motifs on sequence memory and recall. We test the learning of these abstract motifs in a much longer sequences than previous work, demanding participants to gradually build up their knowledge of the motif while learning.

We study the effect of memorizing projectional and variable motifs in sequences by asking the following questions: 1. Are sequences constructed according to an underlying motif memorized more accurately than randomly generated sequences, and 2. Are novel sequences sharing the same motif recalled more accurately than random sequences? We ask these questions in two experiments, each studying one proposed motif type. Furthermore, we hypothesized that identifying structures as motifs helps to simply memory representations of long sequences. We implemented this assumption in our computational model, which continuously finds recurring motifs from distinct sequences. The model learns motifs as abstract representations incorporating components from transition probabilities, chunks, and motifs to reduce memory complexity. We look at the learning and transfer abilities of participants in comparison to the model.

## Methods
### A taxonomy of sequence motifs
We define sequence motifs as underlying sequence patterns that are not on the superficial item level but only detectable after performing transformations on sequences of items. We define and study two types of sequence motifs: projectional and variable. An illustration of the two motif types is shown in Fig. 1.

**Projectional motif** refers to a pattern in a projected space shared amongst distinct sequences. Some transformation functions can map the superficial sequential content to a projected space as illustrated in Fig. 1a, a projectional motif denoted as XYYX appears in distinct sequences ACCA and sequence BEEB shares (with X being the first appearing unique item in the sequence, and Y being the second appearing item). In relation to Beethoven's Fifth in the introduction, the music phrases GGGE♭ and FFFD contain a projectional motif XXXY.

**Variable motif** refers to a pattern containing invariant and variant parts of the sequence. A sequence with a variable motif contains at some position a variable—a symbol representing a quantity that can vary in its identity. An example is illustrated in Fig. 1a. The variable "X," described by a gradient-colored box, is an unknown entity representing the possible occurrence of A, C, or E. The same underlying variable motif appears in sequence AXCD and sequence DXFE in the example. They share the same structure of having a varying entity at the location of "X" and unchanging entities at the rest of the sequence positions. In relation to the example in the introduction: in Beethoven's Fifth, a variable motif underlies the music phrase GGGE♭, GGGB, and GGGC, which progresses with the symphony.

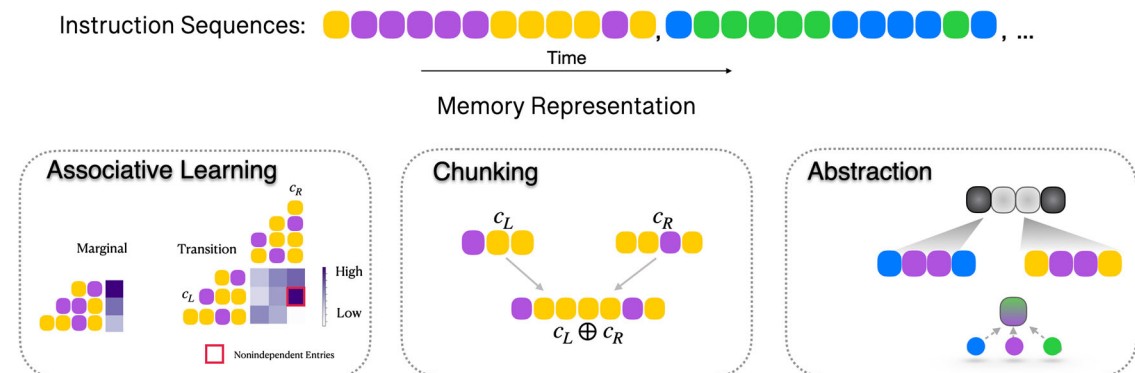

**Fig. 2 | Motif learning model.** Upon observations of instruction sequences, the model acquires the transition frequencies between the learned chunks, combines previously learned chunks into new ones, and looks for abstract representations to compress its sequence memory.

Here, we construct a model that learns abstraction in the case of projectional and variable motifs to reduce representation complexity. The model first tracks the transition probabilities in an abstract space and then gradually chunks sequential elements together. We will test the predictions of our model in two experiments.

### Motif learning model

We put forward a model (Fig. 2) that learns to memorize long sequences via a combination of three strategies: associative learning, chunking, and motif abstraction.

**Associative learning.** When a sequence is presented to the model, the model keeps track of the observational frequencies and the transition frequencies between subsequently presented items. Once an item has been identified, its occurrence frequency will increment by 1, and so will the transition frequency between the current item and the previously identified item. Meanwhile, all frequencies are subject to memory decay via multiplying the count of both the marginal and transition frequency entries by a decay parameter $\theta < 1$.

**Chunking.** Apart from associative learning, the model also remembers sequences by chunking. This part of the model is based on the hierarchical chunking model described in our earlier work (HCM[10]). The model stores learned chunks in long-term memory. These chunks are used in addition to observation and transition frequencies to parse the instruction sequence. The model keeps track of the marginal frequencies of chunks and the transitional frequencies between chunks. A new chunk is created by combining two correlated consecutively occurring chunks into a longer chunk. The combined chunk is then added to the memory of the model. This feature enables the model to learn longer and longer sequences with practice. A picture of how memory chunks are acquired during learning is: at the beginning of the training block, the model stores no sequence segments, and therefore, the model parses the first instruction sequence as 12 sequences of unitary length. These unitary sequential chunks are stored in memory as distinct units. As the model learns to combine previously learned chunks into larger chunks, these larger chunks are, in turn, used to parse the upcoming instruction sequences. During the parsing process, the memory chunk of the largest size, consistent with the upcoming instruction sequence, is identified. In this way, the longer the sequence segments the model has learned, the fewer segments are needed to parse the instruction sequence, and the further the model can predict the sequence. In this way, the model builds up a stable memory representation of sequences over practice by combining pre-existing stable representation of memory sequences in long-term memory[18,19].

We also formalize memory chunks based on their occurrence probabilities, consistent with memory models with memory strength increasing with practice[11]. The lower bound on the number of bits needed to encode this chunk $c$ to be distinguished from other chunks in memory is $-\log_2 P(c)$. The more probable that a chunk occurs in the instruction sequence, the less the memory encoding cost.

**Abstraction.** When the same sequence is presented repeatedly, subparts of the sequence will gradually chunk to combine into bigger sequence segments. However, this process is slow because it requires many repetitions to form chunks. This is especially problematic when the instruction sequence repeats rarely, as each unique sequence has only a small probability in the sequence observation space, and the number of repetitions would have to be increased for the chunking process to build up a memory of the whole sequence. We propose the learning of projectional and variable motifs as two mechanisms to reduce the complexity of memory representations.

**Abstraction via learning projectional motifs.** The model identifies two unique items to describe the sequence and assigns X to the first occurring item and Y to the second item. In this case, X and Y represent separate entities in the projectional motif space. This will be one way that the sequence can be transformed into a lower-dimensional space, in which only two dimensions exist.

Once observational sequences are projected onto a lower dimensional projectional motif space, the model learns the sequence via associative learning and chunking and builds up memory representations of sequences by combining correlated consecutively occurring chunks in the projectional motif space.

For example, upon seeing ACCC, BDDD, and FEEE sequences, the model will map all three sequences onto the same sequence in the projectional motif space: XYYY. Originally, there needed to be six dimensions to describe the observational sequence, each representing the binary indicator of observing each letter. The abstraction process enables all three sequences to be described by the same pattern in an abstract projectional space. Without abstraction, if each of the three sequences occurs uniformly likely, then the minimal encoding length to distinguish between the three subsequences shall be $-\log P(\frac{1}{3})$. But once the projectional motif has been identified, it explains all observational sequences and demands significantly less encoding memory of $-\log P(1)$.

**Abstraction via learning variable motifs.** Under the demand of learning to remember long sequences, an alternative way to compress sequence representation is to learn variables. A variable is an abstract sequence entity that entails a set of concrete sequence entities/chunks. The model identifies the variable identity whenever any of its entailing entities is identified.

The abstraction model discerns variables by analyzing the structure of the transition matrix. Specifically, the model identifies structural patterns within a series of sequential observation chunks that share a common precursor and successor. For instance, if the model observes that entity A

transitions to B, C, and E, and further notes that B, C, and E each transition to F (as reflected in the transition matrix), it will recognize a new variable encompassing B, C, and E. This variable becomes identifiable when any of the elements B, C, or E are detected.

Once a variable entity has been learned, it is parsed and identified as one entity to join forces with associative learning and chunking. In this way, the variable helps the learning agent discover an overarching pattern in the sequence, which would otherwise demand more sequence observations to be learned as separate memory chunks.

The mechanism of variables naturally leads to sequence compression. For example, assume the following subsequences: BADF, BBDF, and BCDF have been observed to occur equally likely; each subsequence demands a minimal encoding complexity of $-\log P(1/3)$. As soon as a variable X is identified to entail A, E, or C, then the chunk BXDF would suffice to explain all three observational instances, and this chunk demands a minimal encoding length of $-\log P(1)$.

The model learns memory pieces by combining chunking and associative learning. On top of that, sequence abstraction processes, including projectional transformation and identifying variables, help the model to locate recurring motifs in the abstract space, capable of explaining a larger number of sequence observations and thereby learning faster and compressing further.

A natural benefit of learning abstract motifs is generalization to novel, unseen sequences sharing the same motif structure. The previously learned projectional or variable motifs can be reused to remember novel sequences, facilitating novel sequence acquisition and compression.

The model predicts that participants looking for the minimal complexity representation to learn sequences should behave in the following ways:
- When there are underlying projectional or variable motifs in the sequence, participants' representation of the sequence shall decrease in complexity when more sequences are presented with the same motif type.
- Participants who benefit from learning motifs from training sequences will exploit their previously learned motif structure.
- In the case of projectional motif, motif structure that has been learned before will be exploited to memorize a novel sequence that has never been observed by participants.
- When participants learn the representation of a variable and extrapolate it as a sequential unit to be combined with the unvarying part of the sequence, the variable as a concept will be reused when novel sequences share the same variable.

We will test these predictions in detail in the following two experiments.

## Ethics statement
Informed consent was obtained from all participants before participation, and the experiments were performed following the relevant guidelines and regulations approved by the ethics committee of the University of Tübingen (Ethik-Kommission an der Medizinischen Fakultät der Eberhard-Karls-Universität und am Universitätsklinikum Tübingen), under the study title: Experimente zum Sequenz- und Belohnungslernen, with application number 701/2020BO.

Participants' data were analyzed anonymously. Upon agreement to participate in the study, they consented to a data protection sheet approved by the data protection officer of the MPG (Datenschutzbeauftragte der MPG, Max-Planck-Gesellschaft zur Förderung der Wissenschaften).

## Paradigm
Specifically, six equally distanced squares are horizontally placed on the display. Each assumes a distinct color: blue, yellow, magenta, red, green, and teal and corresponds to one legitimate key on the keyboard: S, D, F, J, K, and L. Participants were instructed to place their fingers stationarily on these designated keys throughout the task (left index finger on D, left middle

finger on S, left ring finger on A, right index finger on J, right middle finger on K, and right ring finger on L). To control for finger familiarity biases, a random mapping from keyboard position to color is generated for each participant.

Before the start of each trial, all colors were initially covered by dark shades. The sequence was then presented sequentially by revealing each color for 800 ms followed by a brief re-covering of dark shades for another 200 ms before the next display color. The colored sequence was presented in three groups of four, separated by pauses of 800 ms accompanied by the display of a pair of paws, akin to the structure of a three-prose-poem with four words in each prose and pauses in between.

Following the sequence display, participants were prompted to recall the instructed sequence by pressing the corresponding key. Upon the press of each key, the shade covering the corresponding color would disappear and the color would be revealed for 200 ms. At the end of each group, a pair of paws would appear to signify the completion of one subsequence. At the end of the third recall group, participants received immediate feedback on their recall accuracy and recall time which marks the completion of one trial. Participants were instructed to prioritize both speed and accuracy and received a performance-based bonus based on both factors. Before the official trials, participants completed a practice trial to familiarize themselves with the task. There was no preregistration of this study.

## Recruitment of participants
We recruited 135 participants for Experiment 1 from Prolific, an online crowd-sourcing experimental platform. Out of all participants, thirty-seven were female, ninety-eight were male. Participants' ages ranged from 18 to 67, with an average of 32 and a median of 28. The experiment took an average of 45.06 minutes to complete. As compensation, participants received a base pay of £4 and another performance-dependent bonus up to £4. The average hourly pay for the study was £11.60.

We recruited 120 participants for Experiment 2 from Prolific, out of which thirty-four were female, eighty-six were male. Participants' ages ranged from 19 to 63, with an average of 31.2 and a median of 28. The experiment took an average of 47.55 minutes to complete. As compensation, participants received a base pay of £4 and another performance-dependent bonus up to £4. The average hourly pay for the study was £10.89.

Across both experiments, we did not collect participants' race/ethnicity data.

## Payment
For both experiments, participants receive feedback about their trial-wise bonus, which is dependent on a mixture of their sequence recall accuracy and reaction time and is ceiled to the maximum bonus divided by the number of trials. The reaction time bonus becomes the maximum when the recall reaction time is less than 2000 ms, and is set to 0 when the recall reaction time exceeds 10000 ms. For reaction time in the middle, the bonusfast is calculated as $bonusfast = bonusmax - (10000 - trialrt)/(10000 - 2000) \times maxtrialbonus$. In this way, a reaction time between the two limits will yield a steady bonus increase.

The trial-wise bonus for accuracy is calculated as follows: when the recall accuracy is perfect, the bonusacc is set to maxtrialbonus. And when the recall accuracy is below 50%, which corresponds to more than 6 of the recalled sequences in a false order or a false recalled item, then the bonusacc for this trial is set to 0. A recall accuracy in between will yield a bonusacc calculated as $bonusacc = bonus_max \times (trialacc - 0.5)/(1 - 0.5)$.

Finally, the trial bonus is calculated as an average of the reaction time bonus and the recall accuracy bonus $trialbonus = 0.5 \times bonusfast + 0.5 \times bonusacc$.

At the end of the experiment, trial-wise performance-dependent bonus was summed up to the total amount of bonus that participants will receive.

## Filtering
We applied the same filtering criteria on the training blocks for all groups as a basis to exclude participants: mean reaction time < 10,000 ms (that is

10 seconds to press a sequence of 12 made of two distinct colors), mean recall accuracy ≥ 50%. On top of that, we measured whether participants were learning by inspecting reaction time decrease, as a violation of a decrease in rt would be an indication of distraction during the study. Data distribution was assumed to be normal but this was not formally tested. When applying a linear regression model regressing trial number on reaction time on participant's data during the training blocks, the reaction time should on average, decrease, which translates to having a significant ($p < 0.05$) of a negative beta coefficient. No filtering criteria were applied to the transfer blocks. Filtering excludes 21.4% (29) of participants out of 135. After filtering, 37 participants are left in group m1, 41 in m2, and 28 in group independent. The average accuracy was $0.80 \pm 0.22$, and the average reaction time was $5446 \pm 3723$ ms.

For experiment 2, we excluded participants who took on average more than 20 seconds to recall a sequence during the training block (since experiment 2 employs more colors than experiment 1, we also relaxed this exclusion criteria accordingly). Since the motif condition is harder than the control condition, we applied different exclusion criteria for the two groups, and excluded participants with an average sequence recall accuracy below 50% in the fixed group (as they have to recall the same sequence repeatedly), and below 20% in the variable motif group. Additionally, we excluded people who do not have a significant reaction time decrease ($p < 0.05$) during the training block—an indicator of not learning during the task. The exclusion criteria apply only to the training blocks and no participants are excluded based on their transfer block performance. 23 participants were excluded given that they have violated any of the above-mentioned criteria. After exclusion, 45 participants out of 120 remained in group m1, and 52 remained in group control. The average accuracy was $0.70 \pm 0.28$, and the average reaction time was $8094 \pm 6209$ ms.

**Sequence recall.** The model receives the same instruction sequences to participants as its training sequences, except that the middle pauses were removed. To recall, the initial item of the sequence is used as a primer for the model to recall subsequent sequential items. Based on the sequence segments stored in the model, it samples from the set of sequence segments that are consistent with the sequence prime while giving priority to sampling larger segments. Once the first sequential segment is sampled, the segment becomes the previous item to sample the next segment, which is based on the transition given the occurrence of the previous segment. The recall complexity is evaluated by calculating the sampled probability of the recalled sequence. $P(c_1, c_2, c_3) = P(c_1)P(c_2|c_1)(c_3|c_2)$, calculated from the marginal and conditional frequencies are both stored in the model.

**Random effect structure of regression analysis.** To obtain the maximal random effect structure justified by design without inflating the Type I error rate[20], while balancing the loss of statistical power[21], we systematically select models across multiple possible random effect structures and report the best model that is supported by data. Specifically, when fitting linear mixed effect logistic regression on keypress correctness, we compared across random intercept per participant, random slope per serial position, and trial ID, and always reported the best fitting model that includes any subset of the three random effects.

**Reporting summary**

Further information on research design is available in the Nature Portfolio Reporting Summary linked to this article.

# Results

In Experiment 1, we tested whether people can learn and transfer sequences described by a projectional motif as shown in Fig. 1. In Experiment 2, we tested whether participants remember novel sequences better when these sequences share the same variable structure as shown in Fig. 1.

Taken together, we implemented learning structured motifs as a memory compression strategy in a computational model. The model exhibits similar learning and transfer behavior to participants in two sequence recall experiments testing each motif type.

## Experiment 1: projectional motifs

Experiment 1 tested how projectional motifs could help memorization and transfer by instructing participants to memorize long sequences. In a sequence recall task, participants were instructed to play a memory game and to memorize 12 consecutively displayed colors by a cartoon cat. After the instruction, they had to recall the sequence by pressing the keys corresponding to the colors.

Unbeknownst to the participants, the instruction sequences contained underlying motifs. As shown in Fig. 1, the motifs consisted of two distinct variables, X and Y, and individual motifs were constructed by arranging patterns of Xs and Ys. All sequences contained an equal amount of 6 Xs and 6 Ys to control for stimulus-specific habituation effects. Each participant was randomly assigned to one of the two motif groups (Motif 1; Motif 2), or to a control group (Independent). Motif 1 followed the pattern XYYY YYXX XXYX, while Motif 2 adhered to the format XXXY YXXY YYYX. In the motif groups, the underlying motif remained consistent across trials. Conversely, in the Independent group, a permutation of 6 Xs and 6 Ys was generated for each trial. The instruction sequences were finalized by mapping X and Y to two distinct colors.

The task was divided into training and transfer blocks. The training block comprised 40 trials, after which participants proceeded to three randomly ordered transfer blocks, each testing for Motif 1, Motif 2, and the Independent sequences with 8 trials. The transfer phase occurs immediately following the training phase without explicit notification. In all trials, participants were instructed to recall sequences by consecutively typing keyboard keys corresponding to the displayed item until the length of the instruction sequence was reached. Within a trial, the response of individual key presses is recorded. The number of key-press errors is calculated by evaluating the hamming distance (the minimum number of substitutions required to change one string into the other) between the recalled sequence and the instruction sequence. The trial-recall accuracy was calculated by evaluating the proportion of positions at which the corresponding keys are the same. After participants finish recall, the trial-wise accuracy is displayed in addition to the bonus corresponding to the current trial. Participants are not informed about their specific mistakes or the position where they have made the mistake. To ensure that no sequences in transfer blocks appeared in the training block, the six colors were divided into two sets: the training set with four colors and the transfer set with the remaining two colors.

### Model prediction

Reducing representation complexity through projectional motifs. In the case of projectional motifs, a rational agent that looks for minimal complexity representations shall acquire the unchanging motifs during learning since motifs in the abstract projectional space explain more instances of sequences compared to memorizing concrete sequence instances.

Our hypothesis posits that an underlying motif within training sequences in a projectional space will enhance memory and out-of-distribution transfer. In this context, a sequence of length n can be conceptualized as a point within an n-dimensional space, and out-of-distribution refers to the capacity to transfer the representation to sequences never encountered during training. We anticipate improved learning and memorization performance during training for both motif groups and positive transfer when the two groups are tested on motifs of the same type.

### Training

Behavioral results. We first compared sequence recall accuracy amongst the three groups in the training block as shown in Fig. 3a. We fitted a linear mixed-effects regression model onto participants' trial-wise sequence recall accuracy, assuming a random intercept over participants and excluded trials that were immediate repetitions. We observed a significant effect of group ($\chi^2(2) = 10.85$, $p = 0.004$, Conditional $R^2 = 0.22$), suggesting that participants in the Motif 1 group

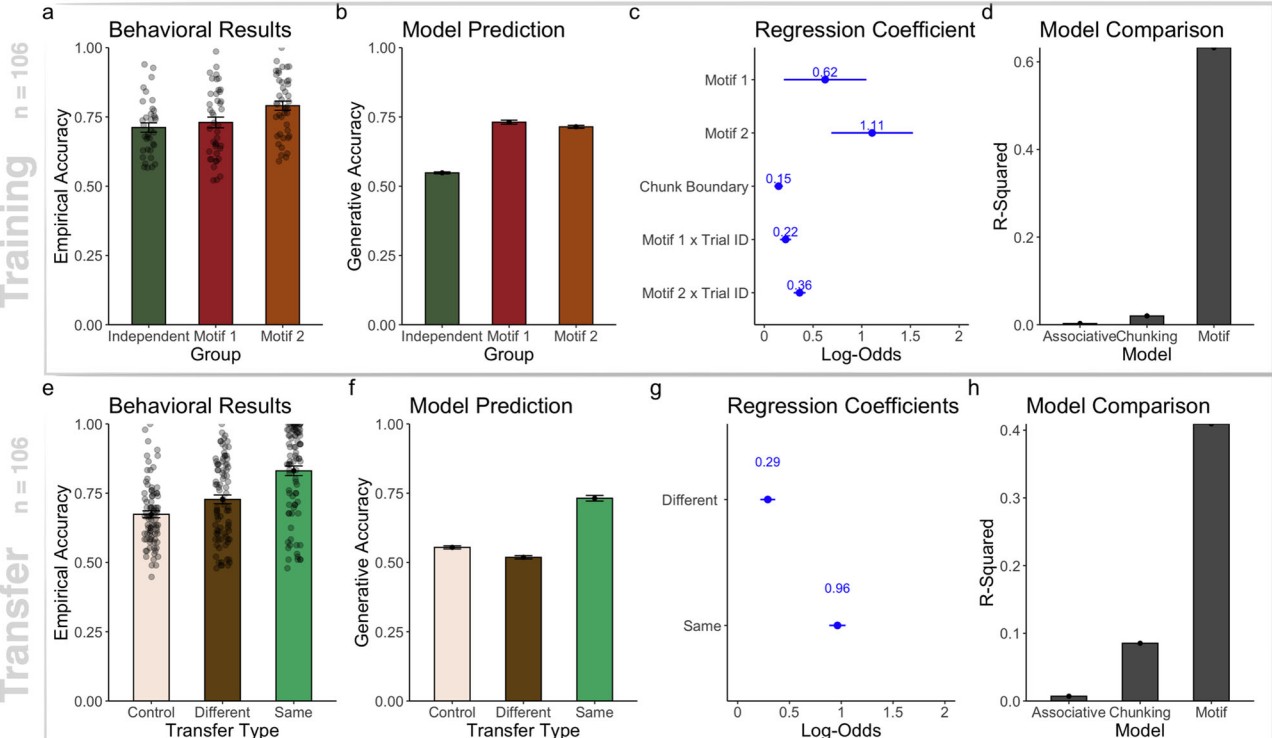

**Fig. 3 | Model simulation and behavioral results for learning and transferring projectional motifs. a** Recall accuracy is higher during the training block in the motif groups than in the control group. **b** Model prediction of sequence recall accuracy training on participants' instruction data. **c** Regression coefficients of the linear mixed-effect model predicting recall accuracy during the training block. **d** Generative accuracy as simulated by the motif learning model correlates with the empirically observed sequence recall accuracy across groups during the training trials. **e** Behavioral results of group-wise recall accuracy across three categories of transfer. Same: Motif 1—Motif 1 and Motif 2—Motif 2; different: Motif 1—Motif 2, and Motif 2—Motif 1; control: Independent—Motif 1, and Independent—Motif 2. **f** Simulation transfer results. **g** Beta coefficients of the logistic regression predicting recall accuracy during the transfer blocks. **h** Correlation between the simulated recall accuracy and participants' recall accuracy.

$(\hat{\beta} = 0.02, se = 0.02, t(109) = 0.7, p = 0.46, 95\% \text{ CI} = -0.03 \text{ to } 0.07)$ and the Motif 2 group $(\hat{\beta} = 0.07, se = 0.02, t(109) = 3.18, p = 0.002, 95\% \text{ CI} = 0.02 \text{ to } 0.13)$ recalled sequences more accurately during the training blocks than those in the Independent group.

**Model simulation.** We compared the behavioral results with the model predictions. We used the same sequences instructed to the participants to train the motif learning model, which creates memory representations of sequence motifs from the observational sequences in an abstract space. We then generated sequences based on the representations learned by the model up to the current time point. We came up with generative accuracy as a surrogate for sequence recall accuracy. The generative accuracy was the edit distance between a generative sequence sampled from the model and the instruction sequence in a particular trial. Figure 3b shows the average generative accuracy of the model. We observed a significant effect of group $(\chi^2(2) = 216.23, p < 0.001, \text{Conditional } R^2 = 0.13)$, suggesting that participants in the Motif 1 group $(\hat{\beta} = 0.18, se = 0.01, t(118) = 22.15, p<0.001, 95\% \text{ CI} = 0.16 \text{ to } 0.20)$ and the Motif 2 group $(\hat{\beta} = 0.17, se = 0.01, t(119) = 20.23, p<0.001, 95\% \text{ CI} = 0.15 \text{ to } 0.18)$ recalled sequences more accurately during the training blocks than the independent group. Similar to participants, the model remembered sequences with underlying motifs more accurately.

**Regression coefficient.** Apart from having higher average recall accuracy, both motif groups improved their recall accuracy faster. As shown in Fig. 3c, we analyzed participants' recall key-press correctness by fitting a logistic regression model assuming a random intercept of each participant and a random slope over individual serial positions (explanation on random effect structure selection in method section Random Effect

Structure of Regression Analysis). We observed an effect for both Motifs (for Motif 1: $\beta = 0.62, se = 0.21, z = 2.88, p = 0.003 \ 95\% \text{ CI} = 0.20 \text{ to } 1.05$; for Motif 2: $\beta = 1.10, se = 0.21, z = 5.17, p < 0.001, 95\% \text{ CI} = 0.69 \text{ to } 1.53$). Apart from that, we observed an interaction effect between the trial number and group $(\chi^2(2) = 51.69, p < 0.001)$. Participants in the Motif 1 group improved their recall accuracy at a faster rate than participants in the Independent group $(\beta = 0.21, se = 0.03, z = 7.55, p < 0.001, 95\% \text{ CI} = 0.16 \text{ to } 0.28)$; the same effect was present for the Motif 2 group $(\beta = 0.36, se = 0.03, z = 11.74, p < 0.001, 95\% \text{ CI} = 0.30 \text{ to } 0.42)$. Thus, people improved faster on remembering sequences with fixed motifs than sequences without.

**Model comparison.** We compared the recall accuracy of the motif learning model with two alternative models: an associative learning model and a chunking model. The motif learning model constructs memory pieces by combining chunking, associative learning, and abstraction via learning projectional motifs. The chunking model contains the same components except for abstraction. The associative learning model learns the first-order transition between observed sequential items. We gave the same instruction sequence to all three models and thereby arrived at an average recall accuracy for each model on each proceeding experimental trial. To do so, we used the same sequences instructed to the participants to train all models. After updating memory components from each trial of sequences, the memory components of the model are used to generate sequences that emulate recall. Then, the model recall accuracy on a particular trial is calculated as the percentage of matching items in the recalled sequence by the models and the instruction sequence. After that, we calculated the group accuracy progression (averaging across participants) for both the model-simulated performance and the participants' performance. The average

generative accuracy per trial of the models is compared to the average recall accuracy per trial of the participants.

We then regressed the generative accuracy of each model onto empirical accuracy and evaluated the goodness of fit by computing the R-squared value. The R-squared measure determines the proportion of variance in the behavioral results that the model prediction can explain and shows how well the data fit the regression model. As shown in Fig. 3d, the motif learning model ($R^2 = 0.63$, 95% CI = 0.55 to 0.71) explained more variance in the behavioral result than a chunking model ($R^2 = 0.02$, 95% CI = 0 to 0.12) that did not abstract. This suggests that abstracting the sequence via projecting the sequence onto the motif space is a critical component that captures human behavior in this task.

Comparing the motif learning model to an associative learning model shows that abstraction alone isn't enough to explain the results. The associative learning model factors in marginal and transition probabilities in the sequences but doesn't learn chunks. Additionally, it explains very little of the variance in human behavior, with $R^2 = 0.003$, 95% CI = 0 to 0.06, compared to the motif learning model. This result suggests that learning the association between items in the projected motif space is insufficient; combining the previously memorized memory chunks into longer memory chunks is also vital to explaining human learning progress.

**Transfer**. We then assessed whether training on motifs affected participants' ability to memorize novel sequences in the transfer blocks.

Behavioral results. We compared participants' performance in the transfer blocks grouped by three transfer types relative to the training block types: Same (Motif 1 - Motif 1, Motif 2 - Motif 2), Different (Motif 1 - Motif 2, Motif 2 - Motif 1), and Control (Independent - Motif 1, Independent - Motif 2). Shown in Fig. 3e, we observed a significant effect of transfer type ($\chi^2(2) = 91.43$, $p < 0.001$, Conditional $R^2 = 0.63$) on recall accuracy. Participants remembered novel sequences with the same motifs more accurately compared to control ($\hat{\beta} = 0.16$, $se = 0.01$, $t(168) = 10.78$, $p<0.001$, 95% CI = 0.13 to 0.19). Surprisingly, we also observed that participants benefited from transferring to a different motif type compared to control ($\hat{\beta} = 0.05$, $se = 0.01$, $t(168) = 3.69$, $p<0.001$, 95% CI = 0.02 to 0.08). Consistent with our hypothesis, training on sequences with motifs helps participants learn novel sequences sharing the same motifs. Participants' reaction time data is also analyzed and visualized in Supplementary References and Figure S1, S2.

Model prediction. Similarly, we evaluated the recall accuracy of the motif learning model on the transfer blocks. Figure 3f shows the generative accuracy of the motif learning model grouped by transfer types. Similar to participants, the model recalled novel sequences with motifs better after it had been trained on the same motif ($\chi^2(2) = 265.43$, $p < 0.001$), compared to having been trained on neither motif ($\beta = 0.18$, $se = 0.01$, $t = 16.69$, $p < 0.001$, 95% CI = 0.16 to 0.2). Different from the participant: it is harder for the model to transfer to an alternative motif type ($\beta = -0.04$, $se = 0.01$, $t = -3.35$, $p < 0.001$, 95% CI = -0.06 to -0.02) than the control. We inspect this discrepancy further in the discussion section.

Regression coefficients. We looked at participants' correctness of recall key presses by fitting a logistic regression model, assuming a random intercept of participants and random slope over individual serial positions and trial numbers (Fig. 3g). We found that the transfer types affect the recall key press correctness ($\chi^2(2) = 679.46$, $p < 0.001$, Conditional $R^2 = 0.23$). Participants who have been tested on the same motif as they had been trained on (m1–m1 and m2–m2) ($\beta = 0.96$, $\sigma = 0.04$, $z = 24.16$, $p < 0.001$, 95% CI = 0.89 to 1.04) are more likely to recall the correct item compared to control. This result resonates with our linear mixed-effect analysis on recall accuracy. Interestingly, participants tested on a motif different from their training motif also did better than the control ($\beta = 0.29$, $\sigma = 0.04$, $z = 7.90$, $p < 0.001$, 95% CI = 0.22 to 0.36). We discuss the implications of this finding further in the discussion section. Additional regression coefficients that confirm the

practice effect, recency effect, and chunk boundary effect are reported in the Supplementary Reference file.

Model comparison. We then compared the resemblance to human behavior between the motif learning model, the associative learning model, and the chunking model (Fig. 3h) during the transfer blocks. Since all three models change their representation when the training schedule switches from training to the transfer blocks, we can compare the generative accuracy of the models to participant recall accuracy. This feature allows us to regress the generative accuracy of each of the three models onto empirical recall accuracy per transfer trial and evaluate the R-squared of the regression as a goodness-of-fit measure.

The motif learning model ($R^2 = 0.41$, 95% CI = 0.26 to 0.55) explains more variance of participants' transfer performance compared to the chunking model ($R^2 = 0.08$, 95% CI = 0.003 to 0.27), suggesting that projecting sequences in a projected motif space, an abstraction process, is critical to capture human behavior in this task. The motif learning model also explains more variance than the associative learning model ($R^2 = 0.05$, 95% CI = 0 to 0.10). Associative learning only is insufficient to capture participants' transfer behavior.

### Experiment 2: variable motifs
Experiment 2 tested the learning and transfer of variable motifs in the sequence recall paradigm. A training block of 40 trials was followed by a transfer block of 24 trials. Participants were split into two groups: the variable motif group (motif) and the fixed group (control). The variable motif group was instructed to remember sequences with variable motif B X D F, D X B F, F X D B (1). X represents a variable and randomly assumes a letter amongst A, C, and E with equal probability with every occurrence. The fixed group was instructed to remember unchanging sequences assuming the form: B A D F, D C B F, F E D B.

During the test block, both groups were instructed to remember a novel sequence with an embedded variable X: D X B F, F X D B, B X F D. The location and entailment of X were the same as the training sequence with variables, but we changed the fixed part of the sequence.

Similar to Experiment 1, the transfer phase proceeds immediately following the training phase without explicit notification. Sequence recall instruction, accuracy evaluation, and feedback are identical to Experiment 1.

**Model prediction**. We hypothesize that when participants are instructed to memorize sequences with a component that varies, identifying variable entities and memorizing them in conjunction with the unvarying part of the sequence should facilitate transfer. That is, when participants encounter novel sequences sharing the same variable entity but different unvarying parts, they should memorize novel sequences with overlapping variables better compared to the control group.

### Training
Behavioral results. Figure 4a shows the average sequence recall accuracy of the variable motif group and the fixed group. We fitted participants' sequence recall accuracy with a linear mixed-effects regression model, assuming a by-participant random intercept. The result showed a significant effect of group ($\chi^2(1) = 50.012$, $p < 0.001$, Conditional $R^2 = 0.42$). The fixed group recalled sequences more accurately than the variable motif group ($\hat{\beta} = -0.22$, $se = 0.03$, $t(95) = -0.806$, $p<0.001$, 95% CI = $-0.28$ to $-0.17$). A changing part of the instruction sequence hindered recall.

Model prediction. We trained the variable motif learning model on the same instruction sequences seen by participants. For sequences with the variable motif, the model learned memory representation manifested in chunks and variables. To do so, the model condensed observations of disparate instances of A, C, and E into one variable entity and concatenates the variable entity with the already-acquired fixed sequence parts in its memory. In this way, the motif learning model learned to represent instruction sequences with variable motifs as a chunk with embedded variable entities. Hence, the

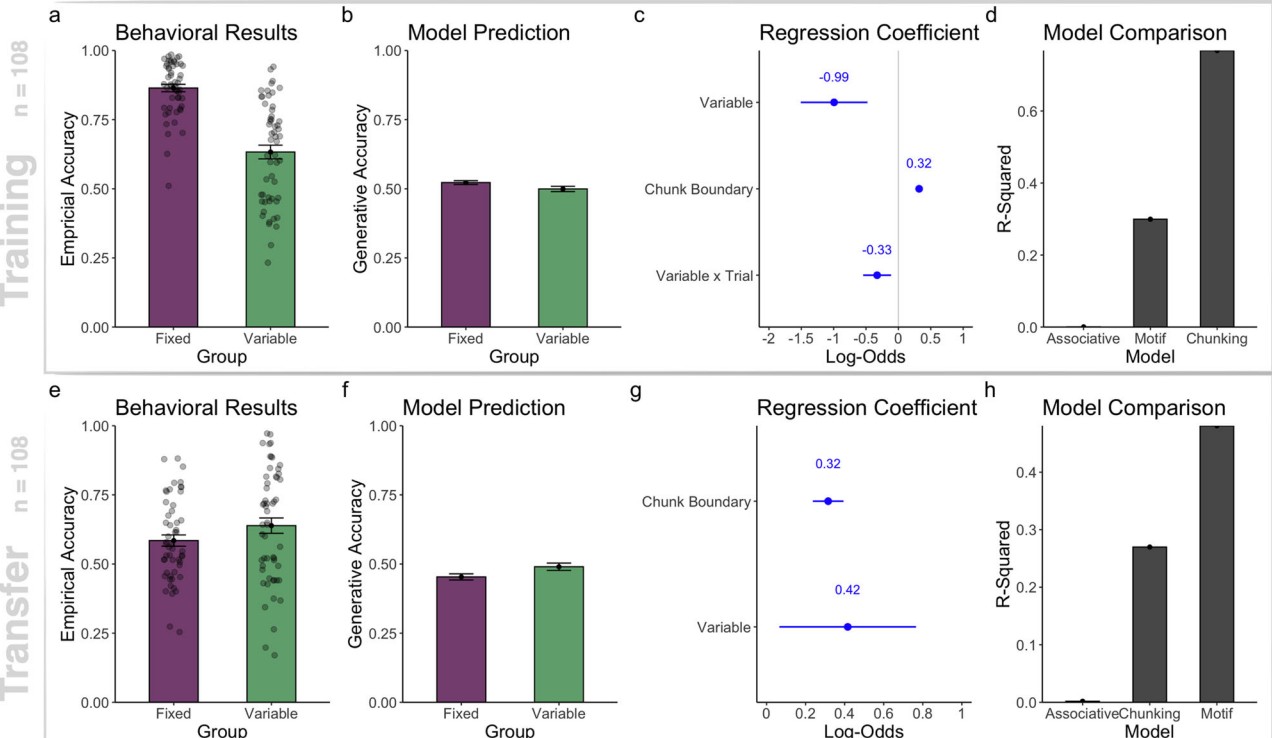

**Fig. 4 | Model simulation and behavioral results for learning and transferring variable motifs. a** Recall accuracy across groups during the training blocks. **b** Simulated recall accuracy during the training blocks. **c** Beta coefficient of a linear mixed effect logistic regression on recall key press correctness during the training blocks. **d** Correlation between simulated generative accuracy and participants' recall accuracy. **e** Recall accuracy across groups during the transfer blocks. **f** Regression coefficients of logistic regression performed on recall keypress correctness during the transfer block. **g** Correlation between simulated transfer generative accuracy and participants' sequence recall accuracy. **h** Correlation between training improvement (average recall accuracy difference between the last five training trials and the first five training trials) and the average recall accuracy during the initial 5 trials of the transfer block.

memory contains both concrete and abstract sequence parts as a low-complexity sequence representation. For control sequences, the model constructed memory pieces by chunking. During recall, sampling entailment chunks of a variable entity introduces memory recall error ($\chi^2(1) = 3.72, p < 0.001$, Conditional $R^2 = 0.03$). The motif learning model recalled sequences with variable motifs less accurately than fixed sequences ($\hat{\beta} = -0.02, se = 0.01, t(106) = -1.93, p<0.05$, 95% CI = -0.04 to 0).

Regression coefficient. We then studied factors that influenced the keypress correctness via fitting a logistic mixed-effects regression, assuming a per-participant random intercept and a random slope per serial position (Conditional $R^2 = 0.32$). Shown in Fig. 4c, the regression coefficient suggested that the variable motif group was more prone to recall mistakes than the fixed group ($\beta = -0.99, se = 0.26, z = -3.78, p < 0.001$, 95% CI = $-1.51$ to -0.48). Apart from that, the variable motif group learned sequences slower than the fixed group ($\beta = -0.33, se = 0.11, z = -2.98, p = 0.002$, 95% CI = $-0.54$ to $-0.11$). Training on sequences with variables decreased participants' probability of recalling the correct key and slowed down learning. Overall, the regression analysis was consistent with our predictions.

Model Comparison. We again compared the motif learning model with an associative learning model and a chunking model by evaluating the R-squared value regressing simulation recall accuracy onto empirical recall accuracy in the same way as in Experiment 1. Figure 4d shows the goodness-of-fit model comparison on the training blocks.

The associative learning model ($R^2 = 0.0005$, 95% CI = 0–0.08) explained very little variance in participants' recall accuracy progression during learning, suggesting that just learning the first-order transition probability was insufficient to explain participants' learning curve on memorizing sequences with variables. Having a chunking component that

builds up recall memory pieces together was essential to explain participants' learning progression. Meanwhile, we observed that the chunking model ($R^2 = 0.76$, 95% CI = 0.65 to 0.86) explained more variance of recall accuracy progression than the variable learning model ($R^2 = 0.39$, 95% CI = 0.14–0.47), possibly because the average chunking process becomes more predictive of participants' recall accuracy than the average variable learning process, as participants may have learned variables in idiosyncratic ways that are not captured by the variable discovery process of the model but are described better by a chunking model.

**Transfer**
Behavioral results. We hypothesized that participants transfer variable representations from the training to the test block. Shown in Fig. 3e is the average recall accuracy of the two groups across all transfer trials. We used an independent-sample t-test to assess the performance difference between the two groups, and a two-sided t-test to assess the superiority of the variable group compared to the fixed group in sequence recall. We observed a significant difference ($t(2317.4) = 4.99; p < 0.001$; $95\% CI = [0.033, 0.076]$) in recall accuracy between the motif group ($M = 0.64$) and the control group ($M = 0.58$), supporting our hypothesis that the variable group performs better at transfer than the fixed group. Participants' reaction time data is also analyzed and visualized in Supplementary References and Figure S3.

Model prediction. As per model simulation shown in Fig. 4f, generative accuracy was higher for the model trained on variable sequences than those trained on fixed sequences ($\hat{\beta} = 0.04, SE = 0.02, t(106) = 2.11, p = 0.03$) ($\chi^2(1) = 4.47, p = 0.03$, Conditional $R^2 = 0.03$). This transfer advantage results from the variable learning model reusing the previously learned variables to parse and chunk in conjunction with the novel sequence part. In other words, the model trained on sequences with variables learned to ignore a

certain part of the novel sequences to afford memorizing the unchanging sequence part.

Regression coefficients. We fitted a mixed-effect logistic regression on participants' recall key press correctness in the transfer block, assuming a per-participant random intercept and a logit link function (Conditional $R^2$ = 0.30). Shown in Fig. 4g, we observed a positive effect of train condition ($\beta = 0.42$, $se = 0.18$, $z = 2.33$, $p = 0.02$, 95% CI = 0.07 to 0.77). Training on sequences with variable motifs helped participants recall novel sequences sharing the same variable motif better than the control group trained on fixed sequences, which was consistent with our model's prediction.

Model comparison. We compared the motif learning model with the chunking and associative learning model on the transfer block. Shown in Fig. 4h, we observed that the motif learning model that reuses its previously learned variables to memorize novel sequences explains the most human recall accuracy variance ($R^2 = 0.48$, 95% CI from 0.33 to 0.65) than the chunking ($R^2 = 0.26$, 95% CI from 0.13 to 0.44) and the associative learning model ($R^2 = 0.001$, 95% CI from 0.0 to 0.16). This aspect suggests that reusing previously learned variables to memorize novel sequences captures a part of the human sequence memory variance when they transfer to novel sequences.

Training improvement correlates with transfer performance. We also assessed the effect of training improvement on transfer performance for both experimental groups. The improvement measure is evaluated on individual participants' sequence average recall accuracy between the last five trials at the end of the training block, subtracted by the first five trials at the beginning of the training block. This difference reflects the average improvement over the training period for every participant. We observed a significant interaction between training improvement and group ($RSS = 2.44$, $F(1) = 10.42$, $p = 0.001$) affecting transfer recall accuracy. Participants who improved more during training on variable motifs performed better during the initial transfer blocks, compared to control ($\beta = 0.53$, $se = 0.17$, $t = 3.22$, $p = 0.002$). Training improvement on variable motifs facilitated transfer to sequences sharing the same variables.

## Discussion

We effortlessly perceive and extract motifs in music, acquire grammatical structure from languages, and use mathematical variables to find out about the unknown. Already during early childhood, we can learn abstract concepts as soon as we learn concrete concepts[22,23]. Linguistics suggest that the conceptual metaphor — mapping similar structural concepts of a known thing to construct an understanding of an unknown concept — plays a vital role in human understanding and reasoning[24,25]. Having seen a solution to a problem, people can solve problems in a similar conceptual relational space[26]. Abstraction as a principle has demonstrated its usage in mathematics and machine learning. Mathematicians have used abstraction as a mapping principle to transfer deductions from one formal system to a new formal system[27]. Abstraction has long been postulated as a crucial requirement for intelligent agents to solve problems in diverse situations[28]. Reinforcement learning studies suggest that state or action abstraction makes the representation more compact, easier to plan, and generalize flexibly to different environments and across tasks[29–33]. Yet, current artificial intelligence systems do not explicitly abstract in the way that humans do[34]. Hence, understanding how humans arrive at abstraction more generically has wide and profound implications in the study of artificial and natural intelligence.

As the key to generalization, transfer, and planning, our ability to abstract from perceptual observations — which has not received sufficient attention relative to its importance in intelligence — urges us to take a closer look at how abstraction arises from sequential perceptual sequences. In the current work, we have proposed two specific sequence abstraction types: projectional motifs — patterns derived from sequences through a projectional function, and variable motifs — patterns that combine both concrete and variable elements. We studied the process of abstract motif learning in sequences, tested the learning and transfer of both motifs in a sequence recall paradigm, and proposed a model that abstracts sequences to compress sequence representations with projectional and variable motifs. We found that our model explained human behavior well.

Previously, associative learning models have been shown to explain human judgment of grammatical versus agrammatical strings in artificial grammar learning tasks[3,4,35,36]. Our model comparison between associative learning and motif learning suggests that associative learning alone is insufficient to explain human abstraction learning and transfer in sequence recall. As an alternative account of sequence learning, chunking models including PARSER[9], HCM[10], CCN and TRACX[11,12] acquire repeated patterns from sequences as chunks. Model comparison between the chunking model and motif learning model suggests that the chunking model captures a part of variable motif learning but not variable motif transfer, nor the learning and transfer of projectional motifs. Expanding the space of chunking from concrete sequences to abstract spaces is vital to capture the motif learning and transfer effects observed in our experiments. In experiment 1, during training, when memorizing sequences with projectional motifs, the chunking model does not align with our observation of human behavior because the model learns chunks on the surface value of the sequences. In comparison, the motif learning model learns chunks in the projectional space of the motifs. While both models learn chunks by a merging mechanism that combines preexisting memorized sequence sub-parts into novel chunks of memorized sequence sub-parts, this chunk-building efficiency correlates with the number of repetitions of the memorized sequence chunks. The motif learning model, having memorized chunks in the motif space, has more opportunities to hone in its memory thanks to the frequent repetition of sequences in the projectional motif space. This model comparison suggests that humans facing this task exert learning behavior that resembles memorization in the projectional motif space rather than memorization of the concrete sequence space.

The inflexibility of memorizing subsequences on the surface value further disadvantages the chunking model in this experiment's transfer phase. Although the chunking model might have learned to compress sequences in chunks in the training phase, the fact that the memory chunks lie in the concrete sequence spaces makes the model inflexible to transfer any learned chunks to the transfer sequences. In comparison, the motif learning model learns chunks in the projectional motif space, which is shared between training and transfer.

In the variable motif learning experiment, the chunking model explains better than the motif learning model on learning variable motifs during the training phase but not the transfer to new ones. This is possible because learning chunks on concrete sequences captures a part of the participant's learning behavior. However, during transfer, the concrete sequence chunks are too stiff to adapt to novel sequences sharing the same variables. The model comparison suggests that the ability of the variable motif learning model to recycle the variable as an entity to construct new chunks is critical to capture the transfer behavior for humans in experiment 2.

## Related Work

A range of cognitive tasks examine learning of surface example structure in text strings. In the artificial grammar learning paradigm, participants learn a subset of the grammatical sequences generated from finite state languages[1]. After observation, they are asked to discriminate grammatical versus ungrammatical (inconsistent with the finite state language) sequences in a test phase. It was observed that participants can generally identify grammatical sequences in the test phase with above-chance accuracy[2]. Previous modeling work suggests that learning the associative transition probabilities between items in the string can replicate participants' performance in the task[3,4,35,36]. Our model comparison between associative learning and motif learning suggests that associative learning alone cannot explain human abstraction learning and transfer in sequence recall.

On top of the first-order transition structure, past research also suggests that people learn explicit structures as frequently occurring fragments

in sequences. Literature suggests that the similarity between the test and training strings influences test judgment[5]. Specifically, test strings that contain overlapping string fragments with the training string are more likely to be judged as grammatical[6–8]. This phenomenon can be explained by chunking models such as PARSER[9], HCM[10], CCN and TRACX[11,12], which learn repeated patterns from sequences as chunks. Our analysis suggests that although the chunking model resembles participants' learning progression during variable motif learning, it fails to capture variable motif transfer or the learning and transfer of projectional motifs. Expanding the space of chunking from concrete sequences to abstract spaces is vital in capturing the motif learning and transfer effects observed in our experiments.

Other works that relate sequence learning and mental compression have used an outlier detection task[37]: participants detect violation upon hearing a binary auditory sequence. Past work has shown that a language-of-thought model's minimal description length of binary sequences relates to human psychological complexity[37,38]. A sequence recall task differs from an outlier detection task in that it directly probes human ability to recall the sequences to be memorized and, therefore, is not limited to testing human prediction of the subsequent element. Our work further relates mental compression with sequence motif learning. Rather than a static account of sequence complexity, the abstraction learning model proposes a discovery process of actively constructing sequence motifs during practice.

The learning of explicit rules bridges between literature and has also been considered in the field of category learning[39]. After presenting participants with observation instances of artificial objects coming from artificially defined category memberships, participants categorize novel objects as belonging to one category or the other. It was observed that both rules and statistics of the categories influence judgment, as atypical examples take longer to be categorized[40,41]. Most theories of rule-based category learning assume that rules and similarities operate on the level of explicit perceptual representations. Relating to our work, we suggest that regularities from observational examples can also be manifested on an even more abstract level, such as the variability structure or projectional space. Assuming that the rule-discovery process operates on projected representations, previous rule-based models are similar to our currently put-forward motif-learning model. If we assume that models originally thought to operate on perceptual representations can also operate on projected representations, different interpretations of the current results become possible. For example, if every presented sequence is stored in abstract space, motifs could also be considered as prototypical abstract sequences[42,43]. Similarly, motifs could also be considered as assemblies of similar multidimensional abstract exemplars[44]. Our results cannot contribute to the debate about people's strategy to learn categories and regularities. The motif learning model is compatible with any of these strategies. However, the current results show that people can transform and use representations in an abstract, projected space to detect regularities, over and above the algebraic rules put forward previously[13] (see Experiment 2).

On the level of learning non-explicit sequence patterns: previous work[13] showed that seven-month-old children could extract an abstract rule when exposed to sequences with simple grammar (e.g., ABA). After exposure, the infants were more likely to direct their gaze toward novel sequences sharing the same structure, such as KTK, rather than toward a different structure, such as DDF. Our experiment further examines the implication of learning projectional motifs in sequence memorization and recall.

The notion of learning variable motifs relates to the symbolic acquisition of language knowledge[14,15], endorsing the view that occurrence frequency cannot be the only basis of grammatical or syntactical language learning, as we can judge very unlikely-occurring sentences to be grammatical[17]. Language acquisition involves learning phase structure, such as a noun phrase usually consistent with a determinant followed by a noun[16]. suggests that abstract patterns on the level of symbols, such as nouns and verbs, operate to utter grammatically valid sentences without an enlisted preoccupied output. The acquisition and utterance of language structure involves the acquisition of operations on the symbolic level.

Previous work has postulated that similarities and rule knowledge are two ends of the same continuum and may have separated learning origins[45]. Moreover, abstraction learning tends to occur after learning the surface-level structure[46]. Perceptual and abstract properties can concurrently occur during the learning process[47]. Our model captures the process of bimodal learning by learning both the surface-level fragments and the deep-level structure and demonstrates its resemblance to human behavior in a sequence recall task that both associative and chunk learning fall short of explaining.

## Limitations

Our work has limitations. In Experiment 1, learning one motif facilitated participants' transfer to a different motif (3e). The same was not true for the model: learning one motif impaired its ability to transfer to the other different motif. The model's ability to recall a new motif is hindered when it has already learned one motif. This occurs because the recall process involves sampling subsequences acquired since the start of training, and the previously learned chunks from the training motif may still get sampled during the recall process which interferes with recall accuracy. This effect is consistent with the proactive interference effect in the literature that memory for previously presented lists impairs memory for later presented lists[48–51]. In contrast, in our experiment, it seems as if humans are establishing a fresh context for structure discovery when encountering a new motif which is not captured by the current model[52–54]. This phenomenon can be attributed to yet an additional layer of contextual abstraction that the model does not capture. Namely, training on sequences with motifs guides people to look for motifs in subsequent sequences. This observation that structural prior prime participants to search for structure in another form resonates with previous findings on structured multi-armed bandit tasks, where a learning-to-learn effect was observed[55]. Future work could extend the current modeling framework to accommodate the flexibility of transferring across motifs. For example, one option would be to introduce a mechanism that updates the prior about the probability of having underlying structure in the sequence. And consequentially, having trained on a motif helps a model to update the structural prior and infer an alternative structural form with a higher likelihood than no structures in the sequence.

In this work, we compared model fit via generative accuracy, which reflects the model's internal memory representation acquired from instruction sequences up to trial n, as it is evaluated on the recalled sequence generated by the model in comparison to the instruction sequence presented on that trial. This method provides one aspect of model fit. Future work could look at other aspects of behavioral-model comparison. One example could be to evaluate the likelihood of participants' recalled sequence given the models and compare the likelihood as a measure of model fit. Alternatively, the complexity of participant generated sequences as parsed by the models can be compared with reaction time data, as less complex sequences would be recalled faster.

Finally, most of our analysis compare model predictions with human behavior on an aggregated level. We encourage future investigations to examine participants' idiosyncratic learning and transfer strategies. Apart from that, our work defines and investigates two particular types of abstraction. We encourage future work to extend the investigation and look at more forms of abstraction or automatic ways of discovering abstraction such as hierarchical clustering and chunking on recursive abstract levels.

## Conclusion

A vital role of abstraction is to facilitate sequence compression and generalization, and we proposed a motif learning model based on this principle. Our model builds up a sequence memory via chunking motifs in an abstract space in search of a low-complexity sequence representation, facilitating memorization and transfer. We developed a sequence recall task to examine whether the two proposed motif types aid in learning and generalization. Our findings suggest that both motifs facilitate sequence memorization and generalization to novel, unseen sequences. Humans showed similar behavior to the model in learning and generalization of both abstraction types.

This suggests that sequence compression via abstraction is a plausible mechanism to explain human performance in sequence memory tasks. Our work paves the way for a better understanding of how people construct abstract representations from observational sequences for efficient compression and transfer.

## Data availability

The data collected is also available at: https://github.com/swu32/motif_learning.

## Code availability

The data collected and code used for analyzing this study can be found in this github repository: https://github.com/swu32/motif_learning.

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

## Acknowledgements

We thank Peter Dayan, Felix Wichmann, and Susanne Haridi for helpful discussions. This work was supported by the Max Planck Society. The funders had no role in study design, data collection and analysis, decision to publish or preparation of the manuscript.

## Author contributions

Conceptualization: Shuchen Wu, Mirko Thalmann, Eric Schulz. Formal analysis: Shuchen Wu, Mirko Thalmann. Software: Shuchen Wu. Visualization: Shuchen Wu. Writing - original draft: Shuchen Wu. Writing - review & editing: Shuchen Wu, Mirko Thalmann, Eric Schulz.

## Funding

## Competing interests

The authors declare no competing interests.
