## [Transparent Peer Review file · Communications Psychology]

Two Types of Motifs Enhance the Recall and Generalization of Long Sequences

Corresponding Author: Ms Shuchen Wu

Version 0:

Decision Letter:

Dear Ms Wu,

Thank you for your patience during the peer-review process. Your manuscript titled "Motif Learning Facilitates Sequence Memorization and Generalization" has now been seen by 3 reviewers, and I include their comments at the end of this message. They find your work of interest but raised some important points. We are interested in the possibility of publishing your study in Communications Psychology, but would like to consider your responses to these concerns and assess a revised manuscript before we make a final decision on publication.

We therefore invite you to revise and resubmit your manuscript, along with a point-by-point response to the reviewers. Please highlight all changes in the manuscript text file.

Editorially, we consider it a necessity to perform additional analyses that comprehensively address all the reviewers' concerns, to improve the presentation, and to avoid unwarranted novelty claims.

I am attaching an Editorial Requests Table that details critical reporting requirements for the revised manuscript. Please attend to each item and ensure your manuscript is fully compliant. We are requesting that your manuscript aligns with these requirements as this facilitates the evaluation of your manuscript, reducing delays in re-review and potential future acceptance. If your revised manuscript is not aligned with these requests on major issues, such as those concerning statistics, it may be returned to you for further revisions without re-review. Additional information can be found in our style and formatting guide <https://www.nature.com/documents/commspsychol-style-formatting-guide-accept.pdf>>Communications Psychology formatting guide.

Please use the following link to submit your

- revised manuscript,
- point-by-point response to the referees' comments,
- cover letter (as a separate document),
- the Editorial Policy Checklist (see below),
- the Reporting Summary (see below), and
- the completed Editorial Request Table (attached):

Link Redacted

We hope to receive your revised paper within 8 weeks; please let us know if you aren't able to submit it within this time so that we can discuss how best to proceed. If we don't hear from you, and the revision process takes significantly longer, we may close your file. In this event, we will still be happy to reconsider your paper at a later date, provided it still presents a

significant contribution to the literature at that stage.

We would appreciate it if you could keep us informed about an estimated timescale for resubmission, to facilitate our planning. Please note that going forward, Jennifer Bellintier (jennifer.bellintier@nature.com) will handle your work, please contact her with any questions.

Best regards,

Antonia Eisenkoeck

Antonia Eisenkoeck
Senior Editor
Communications Psychology

&
Jennifer Bellintier
Senior Editor
Communications Psychology

REVIEWER EXPERTISE:

Reviewer #1: abstraction; computational modelling

Reviewer #2: learning; computational modelling

Reviewer #3: learning

REVIEWER REPORTS:

Reviewer #1 (Remarks to the Author):

The manuscript presents results from two online behavioral experiments in which participants were trained either on “projectional” motifs (i.e., sequences that follow the same underlying algebraic patterns but are instantiated via different items, such as ABBB and CDDD) or “variable” motifs (i.e., if I understood correctly, patterns that vary in all elements beside three that are presented in a fixed order such as in DAFG, LBWR, TCOP, where A – B – C forms a subsequence of items that always appear in these same positions and in this same order for following motifs). The results show that participants can learn such motifs and can also remember better new motifs that follow these same patterns. Crucially, associative or chunking models trained on these same motifs fail to account for participants’ performance, whereas a motif learning model that includes a phase of associative learning, a phase of chunking and a phase of abstraction provides a better account of human results.

I have one major concern about the paper, and it pertains its novelty and claims. The paper shows that human adults can learn the underlying program of simple motifs (experiment 1), but this result is already well established in the literature. Also, they say that participants can transfer such learning but I have trouble understanding what they mean by transfer when the same motif is presented again in the testing phase. In fact, in the training phase participants were already presented with motifs varying in colors, so what’s the difference with the transfer? The only actual transfer I see is when participants are trained on one motif AND, in the transfer phase, a different motif is presented and, indeed, results show that participants can “transfer” their ability/training to look for structures to a different, novel pattern, as compared with the control condition in which no regular pattern was presented during training (and thus no attention towards structure could be present).

Experiment 2 is somehow more novel, since it shows the learning of a specific sequence “hidden” in a varying ensemble of items (but I feel it is just showing a new kind of patterns that humans can learn).

Given these first elements, what is really new about the fact that, as expressed in the title, motif learning facilitates sequence memorization? I personally think that the authors should stress much more their model comparison, since this is what be truly novel with respect to previous literature (although I have some comments on these model comparisons, that I detail below). The current emphasis on empirical results seems to me to downplay the modelling part.

Lastly, I find the manuscript somewhat hard to read. The terminology is very hard to follow. Some concepts are repeated multiple times and some other are not explained in sufficient detail (notably, some aspects of the experimental paradigm are not clear, as I explain below).

Specific comments:

- The terminology for the motifs is awful, and unlikely to be adopted in the future literature. The words “projectional” and “variable” motifs do not seem to very adequately capture what is meant. It took me a while to understand what they refer to. A more transparent representation of them in figure 1 would help. Furthermore, using the term “variable” to also indicate a sub-condition of the “variable group” clearly does not improve readability.
- Be careful in the legend of your figure 1: letter d does not refer to what is described in the figure legend.
- Line 62, page 3: “memory representations become less complex”. Please clarify.
- Training and transfer phases of experiments should be much better explained: how did the transfer test work? Did it occur immediately after the training? What was considered as a correct response? Only when all elements were correctly

repeated? Were participants given feedback only about their overall answer or where they informed about the specific mistakes they made?

- I think it would be interesting to also have a look at incorrect trials: how many elements are mis-ordered on average? How do such mistakes differ from the model's mistakes?
- In figure 3 and 4 I suggest to add subtitles such as "training" for upper plots and "transfer" for bottom plots to improve readability.
- Line 350, page 11 "out" should be "our" (?)
- Line 351, page 11. Maybe you should find a clearer word than "gramaticity"?
- Line 357, page 11. I would definitely expand much more the implications and/or authors' opinions on why different models better explain different results. For example, why chunking model alone explains better the learning of variable motifs but not the abstraction to new ones? Would it suggest that chunking is not that "abstract", meaning that it can efficiently compress information but only on concrete sequences, without generalizing? I suggest to develop the discussion of this. In fact, the "motif learning" model is good only for three conditions out of 4 and this might be quite a considerable limitation of the study.
- Could you report results on reaction times?

Reviewer #2 (Remarks to the Author):

Summary: The current paper trains subjects on sequences containing 1 out of 2 abstract principles ("motifs"). A number of models is presented that (also) store sequences based on the same abstract principles. It is shown that sequences obeying the abstract principles are generally better recalled; and that the model incorporating these principles are tend to generate sequences obeying them.

The data is interesting, and the motifs model shows an interesting combination of associative learning, chunking, and abstraction. I have a number of comments, mostly related to detail of reporting.

I had a hard time extracting the meaning of the two motifs from the introduction; I suggest putting some concrete examples (that now appear only in the Results) in the Intro also.

All the models are very briefly described only; there is just a qualitative description scattered across Introduction, Results, and Methods section (I realize this is partially due to the journal format). Minimally, the algorithm of each model should be specified, including all the model's parameters, the values they take on, the robustness of the value settings, and how the values were chosen. Did you also try to estimate the parameters based on the data? If not: Why not? If yes: Can you reliably recover model parameters?

Relatedly, the authors claim they regressed generative onto empirical accuracy: But it is not clear what is the unit of measurement. Is this across trials, and then the average across subjects? Were model parameters optimized in some way? I think the report must be much more explicit on the modeling and how it relates to the data analysis.

Perhaps the most interesting finding is that participants show a transfer between motifs, suggesting (as the authors indeed also note) that the participants go into a "structure discovery mode". Can the authors speculate how the model could be extended to capture this phenomenon?

I don't think generative accuracy is a very convincing measure of model fit, given that it's completely independent of the data on trial n (when data are generated on trial n). I think the authors could acknowledge that and propose in the discussion some better measure for future research.

In experiment 2, why does the variable group perform worse on the training data?

Filtering criteria: please also report the % of participants that is filtered (in or out)

Reviewer #3 (Remarks to the Author):

This paper reports evidence that people learn two types of "motifs" (abstract structures) in a sequence memory task. The first is projectional motifs, which consist of repetitions of sequences of the same relational structure. The second is variable motifs, which are similar except that some elements can be concrete (i.e. fixed) and others can be variable (i.e., varying probabilistically). The authors perform model comparison by comparing variance explained between different nested models and find that a model containing associative learning + chunking + motif learning mechanisms was the best fit to the data. They conclude that people are capable of learning motifs and that this strategy reduces cognitive load, facilitates memory, and generalizes.

The paper was well-written and mostly clear. One of the model's predictions ("different" transfer in Experiment 1) was not supported by the data, and the authors provided a good explanation for this finding. I don't doubt that people can sometimes learn about abstract underlying structures.

Having said that, I do have some concerns.

1. Literature.

I am unfamiliar with the term “motif” as used in this paper. If this is a completely novel study of this phenomenon, then the authors should explicitly state this, or provide references. Regardless, I think the manuscript would be strengthened by making contact with the vast literature on relational and rule learning in category learning and other related tasks (e.g., artificial grammar, sequence learning, see references below). This would enable the reader to see whether and how the results extended beyond the specific type of memorization task used, and enable the authors to clarify the novel contribution of their work (e.g., the variable motifs seemed more novel to me, but I couldn't be sure because of the lack of citations). I appreciate that there are differences between the current task and previous literature, but it's important to note that studies in related fields typically find a mixture of strategies: some people learn abstract/relational rules, while others learn about similarity or exemplars/instances. The implications of the authors' conclusions is that motif learning should occur frequently, effortlessly, and for all individuals (see point 2), which does not align with the literature.

2. Methodology.

The authors state in the Methods section that they excluded participants based on a regression analysis of the RTs in training. If participants did not show an overall decrease in RTs in training, then they were deemed not to have learned and were excluded from subsequent analysis. The authors then compare groups on their learning performance, and their transfer performance. This is a very different approach to what I'm used to. Typically, exclusions are applied to the training data to ensure that participants who do not learn are not included (same as in this study). However, the critical data are the transfer/test data. Here, different test items are presented that allows for diagnosis of whether participants have learned abstract rules/relations or not. So the exclusions are applied to training, but conclusions regarding WHAT participants have learned are drawn on the basis of the test data. The reason this design permits inference about learning of abstractions is because learning in the training phase does not require learning any motifs or abstract rules. But those who do, show different patterns of transfer at test compared to those who do not.

My concern with the methodology in this manuscript is that it is somewhat circular to say that people learn motifs, when you excluded participants who haven't shown evidence of learning motifs. One could also argue that the predictions on page 5 (when participants show evidence of learning motifs in training, they will show evidence of transferring their knowledge) are also circular. The predictions make sense initially, until you realise that the only way that participants can pass the training criterion is by getting faster over trials, which ALREADY suggests they are transferring their knowledge.

Perhaps I am missing something, but I think the pros and cons of the specific methodology should be addressed. At the very least, the authors need to qualify their conclusions by emphasising the implications of the training criterion. For example, the study says nothing about WHEN people learn motifs, just that they can/do. The conclusions are phrased as though this is a common and adaptive form of learning that people should always exhibit, but the fact that motif learning was found is not surprising given the experimental design and exclusion criteria.

Related, it would be useful to state whether the results were the same when no exclusions were applied. Given the unequal number of exclusions in each group, I assume that the results should not change, but it would be good to know for sure.

A final comment on the RT exclusion criterion is that it seemed a bit odd to apply it to the Independent group in Experiment 1, where there was nothing to learn. Sure, there might be practice effects, but there could also be fatigue effects. I guess it's also odd to compare two groups with different exclusion criteria, hence why I don't think the current design is optimal to address the research question.

3. Model comparison.

The authors use R-squared to compare the models, which does not account for model flexibility/complexity. This is important because the models seem to be nested, in that more complex models had more mechanisms, and the most complex model was best-fitting in both experiments. I was surprised that this was not discussed in the limitations. Perhaps I have misunderstood how the motif learning model works, in which case more detail could be given on how the 3 mechanisms of associative learning, chunking, and motif learning interact.

Minor comments

- I don't know what aspect of the results supports the first prediction on page 5 line 140
- I'm not sure the use of one-tailed tests is defensible. They are not appropriate unless an effect in the opposite direction is impossible or uninterpretable.
- “out” should be “our” (page 11 line 350)

References

- Goldwater, M. B. , Don, H. J. , Krusche, M. F. & Livesey, E. J. (2018). Relational Discovery in Category Learning. *Journal of Experimental Psychology: General*, 147 (1), 1-35. doi: 10.1037/xge0000387.
- Little, J. L., & McDaniel, M. A. (2015). Individual differences in category learning: Memorization versus rule abstraction. *Memory & cognition*, 43, 283-297.
- Pothos, E. M. (2005). The rules versus similarity distinction. *Behavioral and brain sciences*, 28(1), 1-14

EDITORIAL POLICIES

We ask that you ensure your manuscript complies with our editorial policies and reporting requirements.

To that end, we require revised manuscripts to be accompanied by two completed items: a reporting summary that collects information on study design and procedure, and an editorial policy checklist that verifies compliance with all required editorial policies.

- <https://www.nature.com/documents/nr-reporting-summary.zip>>Nature Research Reporting Summary
- <https://www.nature.com/documents/nr-editorial-policy-checklist.pdf>>Editorial Policy Checklist

All points on the policy checklist must be addressed. Your revised manuscript can only be sent back to the referees if these checklists are completed and uploaded with the revision.

Notes: If you have submitted a Stage 1 Registered Report, Review, Primer, Comment, or Perspective you do not need to submit these forms. If you have already submitted these forms, you may disregard this request.

If you experience problems in linking your ORCID, please contact the <http://platformsupport.nature.com/> Platform Support Helpdesk.

Version 1:

Decision Letter:

Dear Ms Wu,

Your manuscript titled "Two Types of Motifs Enhance the Recall and Generalization of Long Sequences" has now been seen by our reviewers, whose comments appear below. Please note, the reviewer report #1 you received previously was authored jointly by two experts from the same lab. Only one of these reviewers was able to re-review the work. In light of their advice I am delighted to say that we are happy, in principle, to publish a suitably revised version in Communications Psychology under the open access CC BY license (Creative Commons Attribution v4.0 International License).

We therefore invite you to revise your paper one last time to address the remaining concerns of our reviewers and a list of editorial requests. At the same time we ask that you edit your manuscript to comply with our format requirements and to maximise the accessibility and therefore the impact of your work.

EDITORIAL REQUESTS:

SUBMISSION INFORMATION:

OPEN ACCESS:

Communications Psychology is a fully open access journal. Articles are made freely accessible on publication under a [CC BY license](http://creativecommons.org/licenses/by/4.0) (Creative Commons Attribution 4.0 International License). This license allows maximum dissemination and re-use of open access materials and is preferred by many research funding bodies.

For further information about article processing charges, open access funding, and advice and support from Nature Research, please visit <https://www.nature.com/commspsychol/article-processing-charges>.

At acceptance, you will be provided with instructions for completing this CC BY license on behalf of all authors. This grants us the necessary permissions to publish your paper. Additionally, you will be asked to declare that all required third party permissions have been obtained, and to provide billing information in order to pay the article-processing charge (APC).

* DATA AVAILABILITY:

Link Redacted

Best regards,

Jennifer Bellingtier

Jennifer Bellingtier, PhD
Senior Editor
Communications Psychology

REVIEWERS' EXPERTISE:

Reviewer #1: abstraction; computational modelling
Reviewer #2: learning; computational modelling

REVIEWERS' COMMENTS:

Reviewer #1 (Remarks to the Author):

I congratulate the authors on their efforts. They made a very detailed revision, explaining all their major changes and took into serious considerations all my concerns, that I think have been well addressed. Just a personal comment (or humble suggestion for the future?): the flow of the manuscript is still sometimes a little "hard" to follow. A few examples of complex sentences: 1) "The inflexibility of memorizing subsequences on the surface value further disadvantages the chunking model in this experiment's transfer phase". 2) "The learning of explicit rules bridges between literature". Or some other times sentences are way too long and should be better parsed. Example: "When subjects learn the representation of a variable and extrapolate it as a sequential unit to be combined with the unvarying part of the sequence, the variable as a concept will be reused when novel sequences sharing the same variable but distinct varying sequence structure need to be remembered". The experimental design and concepts are already quite un-intuitive, as I said in my previous review and this is why it becomes particularly important to pay extra attention to the structure and "flow" of the manuscript :)

Reviewer #2 (Remarks to the Author):

The authors have replied to my comments in a satisfactory manner.

1 Reviewer 1

1.1 Overall evaluation

The manuscript presents results from two online behavioral experiments in which participants were trained either on “projectional” motifs (i.e., sequences that follow the same underlying algebraic patterns but are instantiated via different items, such as ABBB and CDDD) or “variable” motifs (i.e., if I understood correctly, patterns that vary in all elements beside three that are presented in a fixed order such as in DAFG, LBWR, TCOP, where A – B – C forms a subsequence of items that always appear in these same positions and in this same order for following motifs). The results show that participants can learn such motifs and can also remember better new motifs that follow these same patterns. Crucially, associative or chunking models trained on these same motifs fail to account for participants’ performance, whereas a motif learning model that includes a phase of associative learning, a phase of chunking, and a phase of abstraction provides a better account of human results.

Thank you for the concise summary, thorough review, and insightful comments on our manuscript. We appreciate your engagement with the experimental design and our study’s core findings.

However, we would like to clarify one major point regarding variable motifs. Variable motifs refer to an underlying pattern of variation, i.e., the specific ordinal position of the sequence contains elements that exhibit variability, while others maintain a fixed order. We picked the name ‘variable motif’ because the embedded variable in a variable motif resembles a mathematical variable.

In maths, a variable is a symbol representing a quantity that can vary or change, often used in expressions, equations, or functions to denote unknowns or placeholders. For example, in $2x + 3 = 7$, x represents an unknown quantity.

Inside the reviewer’s example DAFG, LBWR, and TCOP, the three elements at positions A, B, and C vary, but all other elements remain fixed in the same position and order. Hence, in the illustration in Figure 3, we replaced the occurrence location of the variable with X to highlight that X is a placeholder for an unknown quantity that can happen to be randomly A, B, or C. We will ensure this distinction is clearly articulated in the manuscript to avoid ambiguity. We highlight the changes that we have made in section 1.3.

1.2 Novelty Claims

I have one major concern about the paper, and it pertains its novelty and claims. The paper shows that human adults can learn the underlying program of simple motifs (experiment 1), but this result is already well established in the literature. Also, they say that participants can transfer such learning but I have trouble understanding what they mean by transfer when the same motif is presented again in the testing phase. In fact, in the training phase participants were already presented with motifs varying in colors, so what’s the difference with the transfer? The only actual transfer I see is when participants are trained on one motif AND, in the transfer

phase, a different motif is presented and, indeed, results show that participants can “transfer” their ability/training to look for structures to a different, novel pattern, as compared with the control condition in which no regular pattern was presented during training (and thus no attention towards structure could be present). Experiment 2 is somehow more novel, since it shows the learning of a specific sequence “hidden” in a varying ensemble of items (but I feel it is just showing a new kind of patterns that humans can learn).

We thank the reviewer for raising this concern. We agree that some previous studies have suggested that people are sensitive to "the underlying program of simple motifs" that resemble projectional motifs. However, we would like to point out that studies such as G. F. Marcus, Vijayan, Bandi Rao, and Vishton (1999) or Gomez and Gerken (1999) focus on testing the transfer ability of similar motifs (AAB, EEF) in the artificial grammar learning paradigm. In such paradigms, participants observe a subset of the grammatical sequences in a training phase and subsequently need to discriminate grammatical versus ungrammatical sequences in the test phase. It was observed that participants were generally able to identify grammatical sequences in the test phase with above-chance accuracy. Moreover, we highlight two major differences between our work and previous work.

One is that we conducted a sequence recall task and experimented on whether motifs help people memorize and actively recall sequences, as familiarity judgments do not guarantee successful sequence recall (Yonelinas, 2002). Another is that we test learning abstraction structures in a much longer sequence than previous work, which usually consisted of sequences of less than five elements. Because this is a challenging sequence to remember, our task demands participants to gradually build up their knowledge of the motif during the learning period. The previous experiments did not put much cognitive demand on actively constructing projectional motifs. We are unaware of previous experiments testing the memorization and recall of long sequences that contain motifs.

To clarify what is in the transfer phase, transfer learning refers to an improvement in a transfer domain where the task has no overlap with the training domain (Weiss, Khoshgoftaar, & Wang, 2016), and only the relation between the training and transfer task shall help the learning model. In this work, the transfer task tests subjects memorization performance on unencountered tasks, in this case, novel sequences that never appeared in the training set. Although participants are presented with sequences made of different colors in the training phase, they are tasked with remembering novel sequences in the transfer phase that never before appeared in their training phase. Therefore, training on sequences with underlying motifs helps participants transfer to novel sequences sharing the same motif.

Nonetheless, we have taken this comment very seriously and adapted our manuscript in the following two ways. 1. We have reduced all novelty claims, as also requested by the editor. 2. We have substantially expanded on our related literature section both in the introduction (see below) and in the discussion (see response to reviewer 3 3.2) part of the paper.

“Literature has suggested our capability to learn multi-faced aspects of sequences. In what is known as grammatical judgment tasks in artificial grammar learning, after familiarizing subjects to a set of grammatically valid sequences generated by a finite state language (Chomsky & Miller, n.d.; Dulany, Carlson, & Dewey, 1984), subjects acquire the ability to distinguish unseen grammatical from ungrammatical sequences Gomez and Gerken

(1999); Gómez (2002). Further research suggested that sequence learning extends beyond learning first-order transition probabilities. Frequently occurring fragments shared between the test and training strings influence test judgment (Brooks, 1992) and are more likely to be judged as grammatical ((Knowlton & Squire, 1996; Knowlton, Squire, & Gluck, 1994; Perruchet & Pacteau, 1990). Such phenomenon can be explained by models that learn repeated sequence fragments as chunks French, Addyman, and Mareschal (2011); Perruchet and Vinter (1998); Servan-Schreiber and Anderson (1990); Wu, Elteto, Dasgupta, and Schulz (2022). Beyond learning sequence fragments and transition probabilities, a few studies suggest the early cognitive capability to acquire sequential patterns on an abstract level: After familiarizing infants early as 7-month-old to sequences such as AAB and CCD, the infants were likelier to direct their gaze toward novel sequences sharing the same structure, such as DDF, rather than a different structure, such as KTK. Such ability to capture what was named as 'abstract algebraic structure' G. F. Marcus et al. (1999) in sequences cannot be explained by learning transition probabilities or chunks. Meanwhile, another abstract pattern has been hypothesized by linguists: we acquire sequence knowledge on a symbolic level Boole (1854); G. Marcus (1995); G. F. Marcus (2001). This ability supports learning phase structures on the level of symbols such as noun phrase = determinant + noun, and helps us to judge the grammaticity of very unlikely-occurring sentences Chomsky (2014).”

We are glad that the reviewer appreciated the novelty of experiment 2. We think it is important to formulate and categorize the kind of patterns that humans can learn and test them in controlled experimental settings. We contribute conceptually by bringing specific notions to the definition of abstraction.

Given these first elements, what is really new about the fact that, as expressed in the title, motif learning facilitates sequence memorization? I personally think that the authors should stress much more their model comparison, since this is what be truly novel with respect to previous literature (although I have some comments on these model comparisons, that I detail below). The current emphasis on empirical results seems to me to downplay the modelling part.

Lastly, I find the manuscript somewhat hard to read. The terminology is very hard to follow. Some concepts are repeated multiple times and some other are not explained in sufficient detail (notably, some aspects of the experimental paradigm are not clear, as I explain below).

We now emphasize the model comparison further in the introduction section of the paper. “We have compared several sequence learning models to explain participants’ training and transfer behavior. The model comparison suggests that expanding the space of chunking from concrete sequences to sequences in a transformed, abstract space for projectional and variable motifs is vital to capture the motif learning and transfer effects for participants. ”

We have made more analogies of the terminology to try to illustrate our point; we address this point in section 1.3.

We have changed our title to "Two Types of Motifs Enhance the Recall and Generalization of Long Sequences".

1.3 Terminology

The terminology for the motifs is awful, and unlikely to be adopted in the future literature. The words “projectional” and “variable” motifs do not seem to very adequately capture what is meant. It took me a while to understand what they refer to. A more transparent representation of them in figure 1 would help. Furthermore, using the term “variable” to also indicate a sub-condition of the “variable group” clearly does not improve readability.

We thank the reviewer for pointing this out. We think that this critique point is partially caused by a misunderstanding of what the motifs are (see also point 1). This was likely caused by us not having stated the definitions clearly. We now explain in much more detail why these motifs have been named this way.

“**Projectional motif** refers to a common pattern in a projected space shared amongst distinct sequences. An example is illustrated in Figure 1a, distinct sequences ACCA and sequence BEEB shares the same underlying projectional motif XYYX (with X being the first unique item that appears in the sequence, and Y being the second). In relation to Beethoven’s Fifth in the introduction, the music phrases GGGE♭ and FFFD share a projectional motif XXXY.”

“**Variable motif** refers to a pattern containing invariant and variant sequential items. A sequence with a variable motif contains at some position a variable — a symbol representing a quantity that can vary in its identity. An example is illustrated in Figure 1a. The variable X, described by a gradient-colored box, is an entity representing the possible occurrence of A, C, or E. The same underlying variable motif appears in sequence AXCD and sequence DXFE in the example. They share the same structure of having a varying entity at the location of X and unchanging entities at the rest of the sequence positions. In relation to the example in the introduction: in Beethoven’s Fifth, a variable motif underlies the music phrase such as GGGE♭, GGGB, and GGGC.”

We have also improved Figure 1 and renamed the "variable group" to the "variable motif group". Additionally, we clarified the motif definition in the figure caption.

1.4 Specific Comments

- Be careful in the legend of your figure 1: letter d does not refer to what is described in the figure legend.

Thank you for pointing out the discrepancy in the legend of Figure 1. We have revised it to ensure accuracy.

- Line 62, page 3: “memory representations become less complex”. Please clarify.

We appreciate your attention to detail regarding the statement on line 62, page 3. We have

Figure 1: Taxonomy of motifs and experimental design. **a.** A taxonomy of sequence motifs. Projectional motifs refer to patterns of sequences in a projected space that are mapped from the concrete sequence space by a projection function. In the example being shown, the projection function finds the distinct items in the sequence and maps sequential observation into a binary sequence $XY YX$, with X being the first unique item appearing and Y being the second. Variable motifs refer to a pattern of invariant (dark box) and variant (gradient-colored box) sequential elements. The variable motif contains at some position a variable — a symbol representing a quantity that can vary in its identity. Such a variable is identified when any of the sequential components it represents is identified. In this example, the variable X represents the possible occurrence of A , C , or E . We hypothesized that participants could learn both types of motifs through practice and exploit their knowledge of both motif types in memorizing and generalizing to novel sequences. **b.** We study motif learning in a sequence recall task. Participants are instructed to remember a sequence of 12 colors displayed one after another in three groups of four items separated by a pair of paws after each group. **c.** Experiment 1 studies learning projectional motifs. Participants are divided into three groups. Two motif groups (Motif 1 and Motif 2) and one control group (Independent). **d.** Each group is first trained on their respective motif or random sequences (Independent) and then tested on randomly interleaved transfer blocks of three types. There are no overlapping sequences between all transfer blocks and training blocks. **e.** Experiment 2 studies learning variable motifs. The variable motif group is trained on sequences with an underlying variable motif. That is, the second position of each subsequence display is randomly drawn among three colors (purple, blue, or green). The fixed group is trained to recall fixed sequences. Both groups are then subsequently tested on novel sequences sharing the variable motif.

revised the text to provide a clearer explanation of how memory representations become less complex.

“We study the effect of memorizing projectional and variable motifs in sequences by asking the following questions: 1. Are sequences constructed according to an underlying motif memorized more accurately than randomly generated sequences, and 2. Are novel sequences sharing the same motif recalled more accurately than random sequences? We ask these questions in two experiments, each studying one proposed motif type. Furthermore, we hypothesized that identifying structures as motifs helps to simplify memory representations of long sequences. We implemented this assumption in our computational model that continuously finds recurring motifs from distinct sequences.”

- Training and transfer phases of experiments should be much better explained: how did the transfer test work? Did it occur immediately after the training? What was considered as a correct response? Only when all elements were correctly repeated? Were participants given feedback only about their overall answer, or were they informed about the specific mistakes they made?

Thank you for your valuable feedback on our experiments’ training and transfer phases. We have provided a more detailed explanation of the transfer test procedures, including the timing, criteria for correct responses, and feedback mechanisms.

“The transfer phase occurs immediately following the training phase without explicit notification. In all trials, participants were instructed to recall sequences by consecutively typing keyboard keys corresponding to the displayed item until the length of the instruction sequence was reached. Within a trial, the response of individual key presses is recorded. The number of key-press errors is calculated by evaluating the hamming distance (the minimum number of substitutions required to change one string into the other) between the recalled sequence and the instruction sequence. The trial-recall accuracy was calculated by evaluating the proportion of positions at which the corresponding keys are the same. After participants finish recall, the trial-wise accuracy is displayed in addition to the bonus corresponding to the current trial. Participants are not informed about their specific mistakes or the position where they have made the mistake. ”

- I think it would be interesting to also have a look at incorrect trials: how many elements are misordered on average? How do such mistakes differ from the model’s mistakes?

Thank you for the comment. We have examined the incorrect trials. We labeled correctly recalled key items and regressed the model recall mistakes on the subject recall mistakes, assuming a random intercept of each participant. We observed an effect for the mistakes in the motif learning model in experiment 1 ($\beta = 0.28$, $se = 0.01$, $z = 16.72$, $p \leq 0.001$) and also in experiment 2: ($\beta = 0.63$, $se = 0.02$, $z = 38.88$, $p \leq 0.001$). For the associative learning model, the effect was weaker in experiment 1 ($\beta = 0.05$, $se = 0.02$, $z = 3.19$, $p \leq 0.001$) and also in experiment 2: ($\beta = 0.55$, $se = 0.01$, $z = 30.89$, $p = 0.02$). A similar effect can be seen for the chunk learning model for experiment 1 ($\beta = 0.08$, $se = 0.02$, $z = 4.85$, $p \leq 0.001$) and also in experiment 2:

($\beta = 0.77$, $se = 0.01$, $z = 46.91$, $p \leq 0.001$), resonating with our model comparison analysis.

- *Could you report results on reaction times?*

Thank you for bringing up the importance of reporting reaction time data. We have added reaction time (RT) analysis to enrich the findings of our study in the supplementary information. We did not include RT analysis in the main text mainly because RT in this task is primarily influenced by the task design, and correlates highly with accuracy, and is much noisier. We treat recall accuracy as a more reliable measure of learning behavior and, therefore, focus our analysis mainly on recall accuracy.

However, we nonetheless appended a reaction time analysis in the supplementary information. While in experiment 1, reaction time analysis results in similar trends and conclusions as in our recall accuracy analysis, in experiment 2, reaction time informs little about subjects' internal representation, but the transfer effect is more implicated in recall accuracy.

“ As shown in Figure 2 average reaction time for the three training groups decreases with practice, and reaction time converges at the end of the training block for all three groups. Figure 2 b shows the reaction time to press the recall sequence within each recall trial. Shown in figure 2 c is the average reaction time across the three training groups. The average reaction time to recall the sequence does not differ significantly amongst the three groups, as indicated via fitting a linear mixed effect regression model onto participants' recall time, assuming a random intercept over individual participants and a random slope over serial positions ($\chi^2(2) = 4.32$, $p = .11$).

Other regressors that showed significant effects are serial position, trial ID, chunk boundary, and repetitions, as shown in Figure 3. Serial position, the n-th item recalled in a trial, affects reaction time. The further the position of a sequence recall, the shorter the reaction time ($\beta = -126.76$, $se = 20.51$, $t = 103.30$, $p \leq 0.001$). Trial ID, i.e., the number of practice trials, also reduces reaction time ($\beta = -117.413$, $se = 5.86$, $t = -20$, $p \leq 0.001$), confirming a practice effect over the training phase. Immediate repetitions of the previous sequence also drives reaction time faster ($\beta = -61.86$, $se = 9.28$, $t = -6.67$, $p \leq 0.001$). Reaction time of the first item in each subsequence position is much higher than other serial positions in the sequence ($\beta = 573.54$, $se = 7.19$, $t = 79.73$, $p \leq 0.001$), reflecting the structure of the task.

Shown in figure 2 d is the average reaction time during the transfer phase: for the groups trained on motifs, transfer type affects their transfer performance ($\chi^2 = 174.05$, $p < 0.001$). When the motif groups transfer to the test blocks, their reaction time to recall the sequence and execute the sequence presses is higher for transferring to the same motif compared to transferring to an independent block ($\hat{\beta} = -109.34$, $se = 9.49$, $t(30417) = -11.514$, $p = < 0.0001$). When the motif group transfers to a different motif, the reaction time speed up is not significantly higher than the transfer to an independent block ($\hat{\beta} = -2.22$, $se = 9.49$, $t(30417) = -0.23$, $p = 0.81$). Having been trained on sequences with motifs, participants recall sequences faster when transferring to a sequence with the same motif but not necessarily to a different motif.

For experiment 2, we also fitted a linear mixed effect regression model onto participants' recall time, assuming a random intercept over individual participants and a random slope over trial ID. As shown in Figure 4, regressors that showed significant effects during the training block are serial position, trial ID, and chunk boundary. Serial position, the n-th item recalled in a trial,

Figure 2: Reaction time analysis. a. Average reaction time across training trials. b. Average reaction time across recall sequence position. c. Average reaction time during the training block. d. Average reaction time during the transfer block across three transfer type. Same: Motif 1 – Motif 1 and Motif 2 – Motif 2; different: Motif 1 – Motif 2, and Motif 2 – Motif 1; control: Independent – Motif 1, and Independent – Motif 2.

affects reaction time. The further the position of a sequence recall, the shorter the reaction time ($\beta = -97.68$, $se = 4.09$, $t = -23.84$, $p \leq 0.001$). Trial ID, i.e., the number of practice trials, also reduces reaction time ($\beta = -134.271$, $se = 19.62$, $t = -6.84$, $p \leq 0.001$), confirming a practice effect over the training phase. Number of repetitions drives reaction time faster ($\beta = -7.15$, $se = 2.23$, $t = -3.20$, $p = 0.002$). Reaction time of the first item in each subsequence position is much higher than other serial positions in the sequence ($\beta = 653.003$, $se = 9.46$, $t = 69.02$, $p \leq 0.001$), reflecting the structure of the task.

During the transfer phase, a linear mixed effect regression on recall time, assuming a random intercept over individual participants and a random slope over trial ID and serial position shows serial position ($\beta = -140.66$, $se = 19.02$, $t = -7.39$, $p \leq 0.001$), and chunk boundary as affecting the reaction time ($\beta = 773.94$, $se = 18.95$, $t = 40.85$, $p \leq 0.001$).

- In figure 3 and 4 I suggest to add subtitles such as “training” for upper plots and “transfer” for bottom plots to improve readability.

We appreciate your suggestion to enhance the readability of figures 3 and 4. We have added subtitles as recommended to clarify the different phases and updated the plots in the rendition of our manuscript.

- Line 350, page 11: “out” should be “our” (?) Thank you for identifying the typo on line 350, page 11. We have corrected it to ensure the accuracy of our manuscript.

Figure 3: Reaction time analysis. a. regression coefficient of experiment 1 on during training. b. transfer

Figure 4: Reaction time analysis. a. regression coefficient of experiment 2 on during training. b. transfer

- Line 351, page 11: *Maybe you should find a clearer word than “grammaticity”?*

Your point about finding a clearer term than “grammaticity” is well-taken. We now use alternative terminology to improve the clarity of our discussion.

“Previously, associative learning models have been shown to explain human judgment of grammatical versus agrammatical strings in artificial grammar learning tasks. Gomez and Gerken (1999); Gómez (2002); Saffran, Johnson, Aslin, and Newport (1999); Saffran, Newport, and Aslin (1996)”

- Line 357, page 11: *I would definitely expand much more the implications and/or authors’ opinions on why different models better explain different results. For example, why does the chunking model alone explain better the learning of variable motifs but not the abstraction to new ones? Would it suggest that chunking is not that “abstract”, meaning that it can efficiently compress information but only on concrete sequences, without generalizing? I suggest developing the discussion of this. In fact, the “motif learning” model is good only for three conditions out of 4 and this might be quite a considerable limitation of the study.*

We appreciate your comments on the implications of our results and the limitations of our study. We have expanded the discussion to address why different models better explain different findings, including the potential implications of these findings for our understanding of chunking, motif learning, and generalization.

“In the variable motif learning experiment, the chunking model explains better than the motif learning model on learning variable motifs during the training phase but not the transfer to new ones. This is possible because learning chunks on concrete sequences captures a part of the subject’s learning behavior. But during transfer, the concrete sequence chunks is too stiff to adapt to novel sequences sharing the same variables. The model comparison suggests that the ability of the variable motif learning model to recycle the variable as an entity to construct new chunks is critical to capture the transfer behavior for humans in experiment 2.

In experiment 1, during training, when memorizing sequences with projectional motifs, the chunking model does not align with our observation of human behavior because the model learns chunks on the surface value of the sequences. In comparison, the motif learning model learns chunks in the projectional space of the motifs. While both models learn chunks by a merging mechanism that combines preexisting memorized sequence subparts into novel chunks of memorized sequence sub-parts, this chunk-building efficiency correlates with the number of repetitions of the memorized sequence chunks. The motif learning model, having memorized chunks in the motif space, ends up having more opportunities to hone in its memory thanks to the frequent repetition of sequences in the projectional motif space. This model comparison suggests that humans facing this task exert learning behavior that resembles memorization in the projectional motif space rather than memorization of the concrete sequence space.

The inflexibility of memorizing subsequences on the surface value further disadvantages the chunking model in this experiment’s transfer phase. Although the chunking model

might have learned to compress sequences in chunks in the training phase, the fact that the memory chunks lie in the concrete sequence spaces makes the model inflexible to transfer any learned chunks to the transfer sequences. In comparison, the motif learning model learns chunks in the projectional motif space, which is shared between training and transfer.”

2 Reviewer 2

2.1 Overall evaluation

Summary: The current paper trains subjects on sequences containing 1 out of 2 abstract principles (“motifs”). A number of models is presented that (also) store sequences based on the same abstract principles. It is shown that sequences obeying the abstract principles are generally better recalled; and that the model incorporating these principles are tend to generate sequences obeying them. The data is interesting, and the motifs model shows an interesting combination of associative learning, chunking, and abstraction. I have a number of comments, mostly related to detail of reporting.

We thank the reviewer for the encouraging comments and the constructive discussion points. Thank you for your detailed feedback and insightful comments on our manuscript. We appreciate your acknowledgment of the interesting findings regarding the motifs model and the transfer phenomenon observed in participants. You can find our detailed responses to every comment below.

2.2 Motif Definition

I had a hard time extracting the meaning of the two motifs from the introduction; I suggest putting some concrete examples (that now appear only in the Results) in the Intro also.

We have added some concrete examples in the Introduction to clarify the meaning of the two motifs.

“**Projectional motif** refers to a common pattern in a projected space shared amongst distinct concrete sequences. An example is illustrated in Figure 1a, distinct sequences ACCA and sequence BEEB shares the same underlying projectional motif XYYX (with X being the first unique item that appears in the sequence, and Y being the second). In relation to Beethoven’s Fifth in the introduction, the music phrases GGGE♭ and FFFD share a projectional motif XXXY.”

“**Variable motif** refers to a pattern containing invariant and variant sequential items. A sequence with a variable motif contains at some position a variable — a symbol representing a quantity that can vary in its identity. An example is illustrated in Figure 1a. The variable

X, described by a gradient-colored box, is an entity representing the possible occurrence of A, C, or E. The same underlying variable motif appears in sequence AXCD and sequence DXFE in the example. They share the same structure of having a varying entity at the location of X and unchanging entities at the rest of the sequence positions. In relation to the example in the introduction: in Beethoven’s Fifth, a variable motif underlies the music phrase such as GGGE_b, GGGB, and GGGC.”

2.3 Model Specification

All the models are very briefly described only; there is just a qualitative description scattered across Introduction, Results, and Methods section (I realize this is partially due to the journal format). Minimally, the algorithm of each model should be specified, including all the model’s parameters, the values they take on, the robustness of the value settings, and how the values were chosen. Did you also try to estimate the parameters based on the data? If not: Why not? If yes: Can you reliably recover model parameters?

We acknowledge the need for a more comprehensive description of the models used in our study, including their algorithms, parameters, and how they were selected. We ensured that this information is provided in a structured manner in the Methods section.

With the parameters, we did not try to estimate parameters based on data because the model is constructed on a descriptive level and was not fit to individual participants. We have added a pseudocode and a more detailed description of the model with the specified parameters in the supplementary information.

Algorithm 1: Motif Learning

Require: *seq*: learning sequences
Require: *cg*: dictionary of learned chunks
Require: *threshold_chunk*: boolean flag for learning chunks
Require: *abstraction*: boolean flag for learning variables

```

1: chunk_record ← {}                                ▷ Initialize chunk record
2: t ← 0
3: while not seq_over do
4:   current_chunks, cg, seq, chunk_record ← identify_latest_chunks(cg, seq)
5:   cg ← learning_and_update(current_chunk, chunk_record, cg, threshold_chunk =
      True)
6:   if abstraction then
7:     cg ← abstraction_update(current_chunks, cg)
8:   end if
9:   cg.forget()                                    ▷ multiple all frequency record by  $\theta$ 
10: end while
11: return cg, chunk_record

```

The model initiates with a dictionary cg , which holds chunks (sub-sequences) and the transition between chunks.

When an instruction sequence is presented to the model, it consecutively parses the sequence via the chunks in the dictionary that contain the biggest size. At each parsing step, the model updates the frequencies of each parsed item and transition frequencies between the previously parsed item and the current one. After parsing a chunk, the boolean flag $thresholdchunk$ and $abstraction$ control the model to create new chunks or to learn new variables.

$thresholdchunk$ is a boolean flag that indicates whether the algorithm will learn and combine new chunks based on the input sequence (True) or just parse the sequence with existing items in the dictionary (False). In case it is true, then the algorithm checks if the currently identified chunk and the previously identified chunk have been conjunctively activated more than a minimum threshold in the transition matrix ($N = 3$). On top of that, a hypothesis test (χ^2) is conducted to assess whether the consecutively parsed chunks are correlated with significance level $p \leq 0.05$. If so, then a new chunk is created by combining the previous with the current and incorporating it into the chunking graph cg . This procedure also includes cases where the current chunk contains variables within.

$abstraction$ is another boolean flag that controls the learning of variables. When this flag is on, the model constructs new variables from chunks that share common ancestors and common descendants, indicating these chunks share similar occurrence contexts. A new variable is created if it connects a set of chunks with a combined frequency above a threshold ($freq_T = 6$).

At the end of each sequence parse, the algorithm performs a "forgetting" step, which multiplies all chunk occurrences and transition frequencies by $\theta = 0.996$.

Abstraction Learning When simulating learning projectional motifs, we turn on the $thresholdchunk$ and learn sequences on the projectional motif space. When simulating learning variable motifs, we set both the $learn$ and $abstraction$ flag to be true.

Chunking When simulating the chunking model, we turn on the $thresholdchunk$ flag and turn off the $abstraction$ flag.

Associative Learning When simulating the associative learning model, we turn off both the $thresholdchunk$ flag and the $abstraction$ flag. Thereby no new chunks are created and the model will learn the transition and occurrence frequencies of the atomic sequential elements.

Recall The recall function simulates the process of sequential recall from a chunk graph, starting with a primed item and proceeding through associative transitions. Given a priming first item of the sequence, the model samples a chunk consistent with the primed first item. Subsequent chunks are sampled based on transition probabilities from the previously recalled chunk (prev). The process repeats until the length of the recalled sequence reaches the desired sequence length $seq_l = 12$.

2.4 Regression Unit

Relatedly, the authors claim they regressed generative onto empirical accuracy: But it is not clear what is the unit of measurement. Is this across trials, and then the average across subjects? Were model parameters optimized in some way? I think the report must be much more explicit on the modeling and how it relates to the data analysis.

We have provided a more explicit explanation of how modeling relates to data analysis, including details on parameter estimation, optimization methods, and the unit of measurement for regression analysis.

“ We compared the behavioral results with the model predictions. We used the same sequences instructed to the participants to train all of the models. After updating memory components from each trial of sequences, the memory components of the model are used to generate sequences that emulate recall. Then, the model recall accuracy on a particular trial is calculated as the percentage of matching items in the recalled sequence by the models and the instruction sequence. After that, we calculated the group accuracy progression (averaging across subjects) for both the model-simulated performance and the participants’ performance. The average generative accuracy per trial of the models is each compared compared with the average recall accuracy per trial of the subjects. ”

2.5 Motif Transfer

Perhaps the most interesting finding is that participants show a transfer between motifs, suggesting (as the authors indeed also note) that the participants go into a “structure discovery mode”. Can the authors speculate how the model could be extended to capture this phenomenon?

We added more discussions on the possible extensions to the model to capture the transfer phenomenon observed in participants and discuss these potential extensions in the Limitation section.

“In contrast, in our experiment, it seems as if humans are establishing a fresh context for structure discovery when encountering a new motif which is not captured by the current model Brown, Neath, and Chater (2007); Dennis and Humphreys (2001); Farrell (2012). This phenomenon can be attributed to yet an additional layer of contextual abstraction that the model does not capture. Namely, training on sequences with motifs guides people to look for motifs in subsequent sequences. This observation that structural prior prime participants to search for structure in another form resonates with previous findings on structured multi-armed bandit tasks, where a learning-to-learn effect was observed (Schulz, Franklin, & Gershman, 2020). Future work could extend the current modeling framework to accommodate the flexibility of transferring across motifs. For example, one option would be to introduce a mechanism that updates the prior about the probability of having underlying structure in the sequence. And consequentially, having trained on a motif helps a model to update the structural prior and infer an alternative structural form with a higher likelihood than no structures in the sequence.”

2.6 Alternative Model Fitting Criterion

I don’t think generative accuracy is a very convincing measure of model fit, given that it’s completely independent of the data on trial n (when data are generated on trial n). I think the authors

could acknowledge that and propose in the discussion some better measures for future research.

We acknowledge that generative accuracy provides only one aspect of the model fit and suggest additional measures for future research in the Discussion section:

“ In this work, we compared model fit via generative accuracy, which reflects the model’s internal memory representation acquired from instruction sequences up to trial n , as it is evaluated on the recalled sequence generated by the model in comparison to the instruction sequence presented on that trial. This method provides one aspect of model fit.

Future work could look at other aspects of behavioral-model comparison. One example could be to evaluate the likelihood of participants’ recalled sequence given the models and compare the likelihood as a measure of model fit. Alternatively, the complexity of participant generated sequences as parsed by the models can be compared with reaction time data, as less complex sequences would be recalled faster. ”

2.7 Others

In experiment 2, why does the variable group perform worse on the training data?

The variable group performs worse on the training data compared to the control group because it is harder to memorize long sequences that change in every trial. For the variable group, the item at the variable position changes in every trial. The control group carries out a much easier task because the sequence is the same in every trial. The varying component in the task makes learning harder for the motif group than just remembering the same sequence over and over as in the control condition.

Filtering criteria: please also report the % of participants that is filtered (in or out)

We now report the percentage of participants filtered in or out based on the filtering criteria used in our analysis.

“Filtering excludes 21.4% (29) of participants out of 135. After filtering, 37 participants are left in group m1, 41 in m2, and 28 in group independent. The average accuracy was 0.80 ± 0.22 , and the average reaction time was $5446 \pm 3723(\text{std})$ ms. ”

“Filtering excludes 22.5% (23) of participants out of 120. After filtering, 45 participants remained in the motif group, and 52 remained in the control group.”

3 Reviewer 3

3.1 Overall evaluation

This paper reports evidence that people learn two types of “motifs” (abstract structures) in a sequence memory task. The first is projectional motifs, which consist of repetitions of sequences of the same relational structure. The second is variable motifs, which are similar except that some elements can be concrete (i.e. fixed) and others can be variable (i.e., varying probabilistically). The authors perform model comparison by comparing variance explained between different nested models and find that a model containing associative learning + chunking + motif learning mechanisms was the best fit to the data. They conclude that people are capable of learning motifs and that this strategy reduces cognitive load, facilitates memory, and generalizes. The paper was well-written and mostly clear. One of the model’s predictions (“different” transfer in Experiment 1) was not supported by the data, and the authors provided a good explanation for this finding. I don’t doubt that people can sometimes learn about abstract underlying structures.

3.2 Clarity of Motifs Definition and Literature Review

I am unfamiliar with the term “motif” as used in this paper. If this is a completely novel study of this phenomenon, then the authors should explicitly state this, or provide references. Regardless, I think the manuscript would be strengthened by making contact with the vast literature on relational and rule learning in category learning and other related tasks (e.g., artificial grammar, sequence learning, see references below). This would enable the reader to see whether and how the results extended beyond the specific type of memorization task used, and enable the authors to clarify the novel contribution of their work (e.g., the variable motifs seemed more novel to me, but I couldn’t be sure because of the lack of citations). I appreciate that there are differences between the current task and previous literature, but it’s important to note that studies in related fields typically find a mixture of strategies: some people learn abstract/relational rules, while others learn about similarity or exemplars/instances. The implications of the authors’ conclusions is that motif learning should occur frequently, effortlessly, and for all individuals (see point 2), which does not align with the literature.

We appreciate Reviewer 3’s comment that our paper was well written and mostly clear. We would like to thank Reviewer 3 for pointing towards the ambiguities related to the newly introduced term "motif" and towards the vast literature on category learning, which bears clear similarities to the current questions.

We do not think that a motif is something completely new that has never been studied before. We therefore added additional paragraphs in the Discussion section contextualizing our new modeling work and the current experiments with previous empirical work and theories, especially of relational rule learning and category learning. We hope that this added paragraph as attached clarifies the relationships between our new work and previous work.

Additionally, there are individual differences in how people may learn motifs. Our experiment and result does not allow us to further pin down, on a cognitive level, what exact strategy people used to learn the motifs. Regarding the claim that motif learning occurs for all individuals, we are not suggesting this; rather, we suggest that motif learning occurs for the average individual in the population. We recognize the idiosyncratic learning curve of the individuals as suggested by the previous literature that some individuals use exemplars more, and others use rule learning more (Little & Mcdaniel, 2014). However, the claim focuses more on the average population-level learning motifs than the control sequences.

“ A range of cognitive tasks examine learning of surface example structure in text strings. In the artificial grammar learning paradigm, participants learn a subset of the grammatical sequences generated from finite state languages (Chomsky & Miller, n.d.). After observation, they are asked to discriminate grammatical versus ungrammatical (inconsistent with the finite state language) sequences in a test phase. It was observed that participants can generally identify grammatical sequences in the test phase with above-chance accuracy (Dulany et al., 1984). Previous modeling work suggests that learning the associative transition probabilities between items in the string can replicate participants’ performance in the task (Gomez & Gerken, 1999; Gómez, 2002; Saffran et al., 1999, 1996). Our model comparison between associative learning and motif learning suggests that associative learning alone cannot explain human abstraction learning and transfer in sequence recall.

On top of the first-order transition structure, past research also suggests that people learn explicit structures as frequently occurring fragments in sequences. People have observed that the similarity between the test and training strings influences test judgment (Brooks, 1992). Specifically, test strings that contain overlapping string fragments with the training string are more likely to be judged as grammatical ((Knowlton & Squire, 1996; Knowlton et al., 1994; Perruchet & Pacteau, 1990). This phenomenon can be explained by chunking models such as PARSER (Perruchet & Vinter, 1998), HCM (Wu et al., 2022), CCN and TRACX (French et al., 2011; Servan-Schreiber & Anderson, 1990), which acquire repeated patterns from sequences as chunks. Our analysis suggests that although the chunking model resembles participants’ learning progression during variable motif learning, it fails to capture variable motif transfer or the learning and transfer of projectional motifs. Expanding the space of chunking from concrete sequences to abstract spaces is vital to capturing the motif learning and transfer effects observed in our experiments.

Other works that relate sequence learning and mental compression have used an outlier detection task(Planton et al., 2021): participants detect violation upon hearing a binary auditory sequence. Past work has shown that a language-of-thought model’s minimal description length of binary sequences relates to human psychological complexity (Dehaene, Al Roumi, Lakretz, Planton, & Sablé-Meyer, 2022; Planton et al., 2021). A sequence recall task differs from an outlier detection task in that it directly probes human ability to recall the sequences to be memorized and, therefore, is not limited to testing human prediction of the subsequent element. Our work further relates mental compression with sequence motif learning. Rather than a static account of sequence complexity, the abstraction learning model proposes a discovery process of actively constructing sequence motifs during practice.

The learning of explicit rules bridges between literature and has also been considered in the field of category learning (Ashby & Townsend, 1986). After presenting participants with obser-

vation instances of artificial objects coming from artificially defined category memberships, participants categorize novel objects as belonging to one category or the other. It was observed that both rules and statistics of the categories influence judgment, as atypical examples take longer to be categorized (Allen & Brooks, 1991; Rips, 1989). Most theories of rule-based category learning assume that rules and similarities operate on the level of explicit perceptual representations. Relating to our work, we suggest that regularities from observational examples can also be manifested on an even more abstract level, such as the variability structure or projectional space. Assuming that the rule-discovery process operates on projected representations, previous rule-based models are similar to our currently put-forward motif-learning model. If we assume that models originally thought to operate on perceptual representations can also operate on projected representations, different interpretations of the current results become possible. For example, if every presented sequence is stored in abstract space, motifs could also be considered as prototypical abstract sequences (Homa, Sterling, & Trepel, 1982; Smith & Minda, 1998). Similarly, motifs could also be considered as assemblies of similar multidimensional abstract exemplars (Nosofsky, 1986). Our results cannot contribute to the debate about people’s strategy to learn categories and regularities. The motif learning model is compatible with any of these strategies. However, the current results show that people can transform and use representations in an abstract, projected space to detect regularities, over and above the algebraic rules put forward previously ((G. F. Marcus et al., 1999), see Experiment 2).

On the level of learning non-explicit sequence patterns: G. F. Marcus et al. (1999) showed that seven-month-old children could extract an abstract rule when exposed to sequences with simple grammar (e.g., ABA). After exposure, the infants were more likely to direct their gaze toward novel sequences sharing the same structure, such as KTK, rather than toward a different structure, such as DDF. Our experiment further examines the implication of learning projectional motifs in sequence memorization and recall.

The notion of learning variable motifs relates to the symbolic acquisition of language knowledge (Boole, 1854; G. F. Marcus, 2001), endorsing the view that occurrence frequency cannot be the only basis of grammatical or syntactical language learning, as we can judge very unlikely-occurring sentences to be grammatical (Chomsky, 2014). Language acquisition involves learning phase structure, such as a noun phrase usually consistent with a determinant followed by a noun. G. Marcus (1995) suggests that abstract patterns on the level of symbols, such as nouns and verbs, operate to utter grammatically valid sentences without an enlisted preoccupied output. The acquisition and utterance of language structure involves the acquisition of operations on the symbolic level.

Previous work has postulated that similarities and rule knowledge are two ends of the same continuum and may have separated learning origins (Pothos, 2005). Moreover, abstraction learning tends to occur after learning the surface-level structure (Goldwater, Don, Krusche, & Livesey, 2018). Perceptual and abstract properties can concurrently occur during the learning process (Shanks & John, 1994). Our model captures the process of bimodal learning by learning both the surface-level fragments and the deep-level structure and demonstrated its resemblance to human behavior in a sequence recall task that both associative and chunk learning fall short to explain.”

3.3 Methodology

The authors state in the Methods section that they excluded participants based on a regression analysis of the RTs in training. If participants did not show an overall decrease in RTs in training, then they were deemed not to have learned and were excluded from subsequent analysis. The authors then compare groups on their learning performance, and their transfer performance. This is a very different approach to what I'm used to. Typically, exclusions are applied to the training data to ensure that participants who do not learn are not included (same as in this study). However, the critical data are the transfer/test data. Here, different test items are presented that allows for diagnosis of whether participants have learned abstract rules/relations or not. So the exclusions are applied to training, but conclusions regarding WHAT participants have learned are drawn on the basis of the test data. The reason this design permits inference about learning of abstractions is because learning in the training phase does not require learning any motifs or abstract rules. But those who do, show different patterns of transfer at test compared to those who do not.

My concern with the methodology in this manuscript is that it is somewhat circular to say that people learn motifs, when you excluded participants who haven't shown evidence of learning motifs. One could also argue that the predictions on page 5 (when participants show evidence of learning motifs in training, they will show evidence of transferring their knowledge) are also circular. The predictions make sense initially, until you realise that the only way that participants can pass the training criterion is by getting faster over trials, which ALREADY suggests they are transferring their knowledge.

Perhaps I am missing something, but I think the pros and cons of the specific methodology should be addressed. At the very least, the authors need to qualify their conclusions by emphasising the implications of the training criterion. For example, the study says nothing about WHEN people learn motifs, just that they can/do. The conclusions are phrased as though this is a common and adaptive form of learning that people should always exhibit, but the fact that motif learning was found is not surprising given the experimental design and exclusion criteria.

Related, it would be useful to state whether the results were the same when no exclusions were applied. Given the unequal number of exclusions in each group, I assume that the results should not change, but it would be good to know for sure.

A final comment on the RT exclusion criterion is that it seemed a bit odd to apply it to the Independent group in Experiment 1, where there was nothing to learn. Sure, there might be practice effects, but there could also be fatigue effects. I guess it's also odd to compare two groups with different exclusion criteria, hence why I don't think the current design is optimal to address the research question.

Thank you for the comment. We understand and sympathize with the reviewer’s concern about circular reasoning based on excluding participants who do not learn. We have also conducted the analysis after removing the exclusion criteria based on RT reduction to check the robustness of our result. We have plotted the updated result in Figure 5 and Figure 6 and examined the regression coefficients. Removing the exclusion criterion resulted in small modifications of the regression coefficients. Nevertheless, the conclusion from the analysis stays intact.

Figure 5: Analysis result of experiment 1 without filtering

The reason we applied this exclusion criteria initially was coming from another standpoint. Because we conducted a crowd-sourcing experiment, there is no insurance over participants using alternative techniques, such as writing the sequence down on a piece of paper to complete the task. Therefore, we wanted to exclude abnormal reaction times that indicate solving the task in a way unintended by the experimenter. As shown in Figure 2 a, reaction time reduces dramatically for the average participant in the task, and this is the case for all groups, including the independent group who do not learn from motifs. From the learning curve, the practice effect dominates quite strongly in reaction time speed-up. Therefore, we have decided that excluding participants based on RT reduction was not a harsh criterion to filter participants who might process an abnormal behavior that may not comply with experimental instruction.

3.4 Model Comparison and Statistical Analysis

The authors use R-squared to compare the models, which does not account for model flexibility/complexity. This is important because the models seem to be nested, in that more complex

Figure 6: Analysis result of experiment 2 without filtering

models had more mechanisms, and the most complex model was best-fitting in both experiments. I was surprised that this was not discussed in the limitations. Perhaps I have misunderstood how the motif learning model works, in which case more detail could be given on how the 3 mechanisms of associative learning, chunking, and motif learning interact.

Thank you for your comment regarding the model comparison and statistical analysis. We do not fit the models to participants' behavior. Instead, models are run generatively and their responses are used to predict behavior, which means that no parameters were fitted, and, therefore, the complexity penalization term is not necessary. We have added additional explanation of our modeling procedure to the supplementary information section and also in response to reviewer 2 2.3.

3.5 Minor Comments

I don't know what aspect of the results supports the first prediction on page 5 line 140.

The prediction translates to observing a stronger practice effect on learning motifs in comparison to having no motifs in the sequence.

“The model predicts that subjects looking for the minimal complexity representation to learn sequences should behave in the following ways:

- When there are underlying projectional or variable motifs in the sequence, subjects' repre-

sensation of the sequence shall decrease in complexity when more sequences are presented with the same motif type.

- Subjects who benefit from learning motifs from training sequences will exploit their previously learned motif structure.
- In the case of projectional motif, motif structure that has been learned before will be exploited to memorize a novel sequence that has never been observed/seen by participants.
- When subjects learn the representation of a variable and extrapolate it as a sequential unit to be combined with the unvarying part of the sequence, the variable as a concept will be reused when novel sequences sharing the same variable but distinct varying sequence structure need to be remembered.

”

I'm not sure the use of one-tailed tests is defensible. They are not appropriate unless an effect in the opposite direction is impossible or uninterpretable.

Thank you for your comment. We have updated the analysis and changed the one-sided t-test to a two-sided t-tests.

“We used an independent-sample t-test to assess the performance difference between the two groups, and a two-sided t-test to assess the superiority of the variable group compared to the fixed group in sequence recall. We observed a significant difference ($t(2317.4) = 4.99$; $p \leq 0.001$; $95\%CI = [0.033, 0.076]$) in recall accuracy between the motif group ($M = 0.64$) and the control group ($M = 0.58$), supporting our hypothesis that the variable group performs better at transfer than the fixed group. ”

“out” should be “our” (page 11 line 350).

We appreciate your attention to detail and thank you for highlighting these minor issues. We have corrected the typographical error on page 11 line 350.

References

- Allen, S. W., & Brooks, L. R. (1991, 1). Specializing the operation of an explicit rule. *Journal of experimental psychology. Learning, memory, and cognition*, *120*, 3-19. doi: 10.1037/0096-3445.120.1.3
- Ashby, F. G., & Townsend, J. T. (1986). Varieties of perceptual independence. *Psychological Review*, *93*(2), 154–179. (Place: US Publisher: American Psychological Association) doi: 10.1037/0033-295X.93.2.154
- Boole, G. (1854). *The laws of thought (1854)*. London,: The Open court publishing company.

- Brooks, L. (1992, 03). Salience of item knowledge in learning artificial grammars. *Journal of Experimental Psychology: Learning, Memory, and Cognition*, *18*, 328-344. doi: 10.1037/0278-7393.18.2.328
- Brown, G., Neath, I., & Chater, N. (2007, 07). A temporal ratio model of memory. *Psychological review*, *114*, 539-76. doi: 10.1037/0033-295X.114.3.539
- Chomsky, N. (2014). *Aspects of the theory of syntax* (No. 11). MIT press.
- Chomsky, N., & Miller, G. A. (n.d.). Finite state languages. , *1*(2), 91-112. doi: 10.1016/S0019-9958(58)90082-2
- Dehaene, S., Al Roumi, F., Lakretz, Y., Planton, S., & Sablé-Meyer, M. (2022, September). Symbols and mental programs: a hypothesis about human singularity. *Trends in Cognitive Sciences*, *26*(9), 751-766. Retrieved 2022-10-16, from <https://linkinghub.elsevier.com/retrieve/pii/S1364661322001413> doi: 10.1016/j.tics.2022.06.010
- Dennis, S., & Humphreys, M. (2001, 04). A context noise model of episodic word recognition. *Psychological review*, *108*, 452-78. doi: 10.1037/0033-295X.108.2.452
- Dulany, D. E., Carlson, R. A., & Dewey, G. I. (1984). A case of syntactical learning and judgment: How conscious and how abstract? *Journal of Experimental Psychology: General*, *113*(4), 541-555. Retrieved from <https://doi.org/10.1037/0096-3445.113.4.541> doi: 10.1037/0096-3445.113.4.541
- Farrell, S. (2012). Temporal clustering and sequencing in short-term memory and episodic memory. *Psychological Review*, *119*(2), 223-271. Retrieved 2023-10-17, from <http://doi.apa.org/getdoi.cfm?doi=10.1037/a0027371> doi: 10.1037/a0027371
- French, R. M., Addyman, C., & Mareschal, D. (2011, October). TRACX: A recognition-based connectionist framework for sequence segmentation and chunk extraction. *Psychological Review*, *118*(4), 614-636. Retrieved 2022-08-13, from <http://doi.apa.org/getdoi.cfm?doi=10.1037/a0025255> doi: 10.1037/a0025255
- Goldwater, M. B., Don, H. J., Krusche, M., & Livesey, E. J. (2018). Relational discovery in category learning. *Journal of Experimental Psychology: General*, *147*, 1-35. Retrieved from <https://api.semanticscholar.org/CorpusID:40562717>
- Gomez, R. L., & Gerken, L. (1999, March). Artificial grammar learning by 1-year-olds leads to specific and abstract knowledge. *Cognition*, *70*(2), 109-135. Retrieved 2022-09-18, from <https://www.sciencedirect.com/science/article/pii/S0010027799000037> doi: 10.1016/S0010-0277(99)00003-7
- Gómez, R. L. (2002, September). Variability and Detection of Invariant Structure. *Psychological Science*, *13*(5), 431-436. Retrieved 2022-09-18, from <https://doi.org/10.1111/1467-9280.00476> (Publisher: SAGE Publications Inc) doi: 10.1111/1467-9280.00476
- Homa, D., Sterling, S., & Trepel, L. (1982, January). Limitations of exemplar-based generalization and the abstraction of categorical information. *Journal of Experimental Psychology: Human Learning and Memory*, *7*(6), 418. Retrieved 2022-01-10, from <https://psycnet.apa.org/fulltext/1982-11301-001.pdf> (Publisher: US: American Psychological Association) doi: 10.1037/0278-7393.7.6.418
- Knowlton, B., & Squire, L. (1996, 01). Artificial grammar learning depends on implicit acquisition of abstract and exemplar-specific information. *Journal of experimental psychology. Learning, memory, and cognition*, *22*, 169-81. doi: 10.1037/0278-7393.22.1.169
- Knowlton, B., Squire, L., & Gluck, M. (1994, 07). Probabilistic classification learning in amnesia.

- Learning memory (Cold Spring Harbor, N.Y.), 1*, 106-20. doi: 10.1101/lm.1.2.106
- Little, J., & McDaniel, M. (2014, 10). Individual differences in category learning: Memorization versus rule abstraction. *Memory cognition*, 43. doi: 10.3758/s13421-014-0475-1
- Marcus, G. (1995, June). Children's overregularization of english plurals: A quantitative analysis. *Journal of child language*, 22(2), 447–459. (Funding Information: [*] I thank Steven Pinker, Fei Xu and two anonymous reviewers for comments on an earlier draft. This research was funded by an NDSE Graduate Fellowship to Marcus, NIH Grant HD 18381 to Steven Pinker (MIT), and grants from NIMH (training grant T32 MH18823) and the McDonnell-Pew Program in Cognitive Neuroscience to MIT's Department of Brain and Cognitive Sciences. Address for correspondence: Gary Marcus, Department of Psychology, Tobin Hall, University of Massachusetts, Amherst, MA 01003, USA. E-mail: marcus@psych.umass.edu.) doi: 10.1017/S0305000900009879
- Marcus, G. F. (2001). The algebraic mind: Integrating connectionism and cognitive science.. Retrieved from <https://api.semanticscholar.org/CorpusID:142639115>
- Marcus, G. F., Vijayan, S., Bandi Rao, S., & Vishton, P. M. (1999, January). Rule learning by seven-month-old infants. *Science (New York, N.Y.)*, 283(5398), 77–80. doi: 10.1126/science.283.5398.77
- Nosofsky, R. M. (1986, July). Attention, similarity, and the identification–categorization relationship. *Journal of Experimental Psychology: General*, 115(1), 39. Retrieved 2021-12-17, from <https://psycnet.apa.org/fulltext/1986-16297-001.pdf> (Publisher: US: American Psychological Association) doi: 10.1037/0096-3445.115.1.39
- Perruchet, P., & Pacteau, C. (1990, 09). Synthetic grammar learning: Implicit rule abstraction or explicit fragmentary knowledge? *Journal of Experimental Psychology: General*, 119, 264-275. doi: 10.1037/0096-3445.119.3.264
- Perruchet, P., & Vinter, A. (1998). Parser: A model for word segmentation. *Journal of Memory and Language*, 39(2), 246 - 263. Retrieved from <http://www.sciencedirect.com/science/article/pii/S0749596X98925761> doi: <https://doi.org/10.1006/jmla.1998.2576>
- Planton, S., Kerkoerle, T. v., Abbih, L., Maheu, M., Meyniel, F., Sigman, M., ... Dehaene, S. (2021, January). A theory of memory for binary sequences: Evidence for a mental compression algorithm in humans. *PLOS Computational Biology*, 17(1), e1008598. Retrieved 2022-12-07, from <https://journals.plos.org/ploscompbiol/article?id=10.1371/journal.pcbi.1008598> (Publisher: Public Library of Science) doi: 10.1371/journal.pcbi.1008598
- Pothos, E. M. (2005, February). The rules versus similarity distinction. *Behavioral and Brain Sciences*, 28(1), 1–14. Retrieved 2024-04-02, from https://www.cambridge.org/core/product/identifier/S0140525X05000014/type/journal_article doi: 10.1017/S0140525X05000014
- Rips, L. J. (1989). *Similarity, typicality, and categorization*. Cambridge University Press.
- Saffran, J. R., Johnson, E. K., Aslin, R. N., & Newport, E. L. (1999). Statistical learning of tone sequences by human infants and adults. *Cognition*, 70(1), 27-52. Retrieved from <https://www.sciencedirect.com/science/article/pii/S0010027798000754> doi: [https://doi.org/10.1016/S0010-0277\(98\)00075-4](https://doi.org/10.1016/S0010-0277(98)00075-4)
- Saffran, J. R., Newport, E. L., & Aslin, R. N. (1996). Word segmentation: The role of distribu-

- tional cues. *Journal of Memory and Language*, 35(4), 606-621. Retrieved from <https://www.sciencedirect.com/science/article/pii/S0749596X96900327> doi: <https://doi.org/10.1006/jmla.1996.0032>
- Schulz, E., Franklin, N. T., & Gershman, S. J. (2020). Finding structure in multi-armed bandits. *Cognitive Psychology*, 119, 101261. Retrieved from <https://www.sciencedirect.com/science/article/pii/S0010028519302518> doi: <https://doi.org/10.1016/j.cogpsych.2019.101261>
- Servan-Schreiber, E., & Anderson, J. (1990, 07). Learning artificial grammars with competitive chunking. *Journal of Experimental Psychology: Learning, Memory, and Cognition*, 16, 592-608. doi: 10.1037/0278-7393.16.4.592
- Shanks, D. R., & John, M. F. S. (1994). Characteristics of dissociable human learning systems. *Behavioral and Brain Sciences*, 17, 367 - 395. Retrieved from <https://api.semanticscholar.org/CorpusID:14849936>
- Smith, J. D., & Minda, J. P. (1998, December). Prototypes in the mist: The early epochs of category learning. *Journal of Experimental Psychology: Learning, Memory, and Cognition*, 24(6), 1411. Retrieved 2022-12-12, from <https://psycnet.apa.org/fulltext/1998-12790-005.pdf> (Publisher: US: American Psychological Association) doi: 10.1037/0278-7393.24.6.1411
- Weiss, K., Khoshgoftaar, T., & Wang, D. (2016, 05). A survey of transfer learning. *Journal of Big Data*, 3. doi: 10.1186/s40537-016-0043-6
- Wu, S., Elteto, N., Dasgupta, I., & Schulz, E. (2022). Learning Structure from the Ground up—Hierarchical Representation Learning by Chunking. In S. Koyejo, S. Mohamed, A. Agarwal, D. Belgrave, K. Cho, & A. Oh (Eds.), *Advances in Neural Information Processing Systems* (Vol. 35, pp. 36706–36721). Curran Associates, Inc. Retrieved from https://proceedings.neurips.cc/paper_files/paper/2022/file/ee5bb72130c332c3d4bf8d231e617506-Paper-Conference.pdf
- Yonelinas, A. P. (2002). The Nature of Recollection and Familiarity: A Review of 30 Years of Research. *Journal of Memory and Language*, 46(3), 441–517. Retrieved from <https://www.sciencedirect.com/science/article/pii/S0749596X02928640> doi: <https://doi.org/10.1006/jmla.2002.2864>